# Spatial and Temporal Constraints on the Composition of Microbial Communities in Subsurface Boreholes of the Edgar Experimental Mine

Patrick H. Thieringer,[a] Alexander S. Honeyman,[a] John R. Spear[a]

[a]Department of Civil and Environmental Engineering, Colorado School of Mines, Golden, Colorado, USA

**ABSTRACT** The deep biosphere hosts uniquely adapted microorganisms overcoming geochemical extremes at significant depths within the crust of the Earth. Attention is required to understand the near subsurface and its continuity with surface systems, where numerous novel microbial members with unique physiological modifications remain to be identified. This surface-subsurface relationship raises key questions about networking of surface hydrology, geochemistry affecting near-subsurface microbial composition, and resiliency of subsurface ecosystems. Here, we apply molecular and geochemical approaches to determine temporal microbial composition and environmental conditions of filtered borehole fluid from the Edgar Experimental Mine (~150 m below the surface) in Idaho Springs, CO. Samples were collected over a 4-year collection period from expandable packers deployed to accumulate fluid in previously drilled boreholes located centimeters to meters apart, revealing temporal evolution of borehole microbiology. Meteoric groundwater feeding boreholes demonstrated variable recharge rates likely due to a complex and undefined fracture system within the host rock. 16S rRNA gene analysis determined that unique microbial communities occupy the four boreholes examined. Two boreholes yielded sequences revealing the presence of *Desulfosporosinus*, *Candidatus Nitrotoga*, and *Chelatococcus* associated with endemic subsurface communities. Two other boreholes presented sequences related to nonsubsurface-originating microbiota. High concentration of sulfate along with detected sulfur reducing and oxidizing microorganisms suggests that sulfur related metabolic strategies are prominent within these near-subsurface boreholes. Overall, results indicate that microbial community composition in the near-subsurface is highly dynamic at very fine spatial scales (<20 cm) within fluid-rock equilibrated boreholes, which additionally supports the role of a relationship for surface geochemical processes infiltrating and influencing subsurface environments.

**IMPORTANCE** The Edgar Experimental Mine, Idaho Springs, CO, provides inexpensive and open access to borehole investigations for subsurface microbiology studies. Understanding how microbial processes in the near subsurface are connected to surface hydrological influences is lacking. Investigating microbial communities of subsurface mine boreholes provides evidence of how geochemical processes are linked to biogeochemical processes within each borehole and the geochemical connectedness and mobility of surface influences. This study details microbial community composition and fluid geochemistry over spatial and temporal scales from boreholes within the Edgar Mine. These findings are relevant to biogeochemistry of near-surface mines, caves, and other voids across planetary terrestrial systems. In addition, this work can lead to understanding how microbial communities relate to both fluid-rock equilibration, and geochemical influences may enhance our understanding of subsurface molecular biological tools that aid mining economic practices to reflect biological signals for lucrative veins in the near subsurface.

Address correspondence to John R. Spear, jspear@mines.edu.

**KEYWORDS** microbial ecology, mining, subsurface microbiology, fluid-rock interaction, geomicrobiology

The intricacies of the deep subsurface have triggered many investigations into the microbial habitability sustaining itself in the rock-hosted subsurface. The chemical drivers that govern the transition from terrestrial surface to subsurface life are not fully understood, especially in near-subsurface zones where the system may behave as a hybrid between surface and subsurface processes. Field sites spanning terrestrial, lacustrine, hot springs, and marine environments have gained traction for molecular biological investigation due to unique, extreme conditions mimicking potential environmental settings of early life on Earth as well as analogs to those on other planetary systems (1–4). While marine systems have unveiled the extent of microbial subsurface habitability in oceanic settings, research aimed at understanding accessible terrestrial environments has gained considerable interest because of more uniquely limiting constraints (low nutrient access, low $O_2$ availability, and elevated temperatures) toward microbiological habitability (5, 6).

The large biodiversity of microbial life in the deep biosphere remains to be explored through terrestrial subsurface investigations (7–11). Terrestrial sites can offer further geologic and hydrologic variability than marine environments due to the presence of unconstrained fluid sourcing and lithology in surface soils and bedrock (7, 12–14). Even more, water-rock interactions demonstrated in terrestrial subsurface environments distinguish the chemical disparity of chemolithotrophic processes that sustain ecosystems of low biomass utilizing restricted sources of energy (15, 16). The contemporary outlook of an "extreme" environment encompasses some variation of a chemical or hydrological endpoint for microorganisms to exist primarily due to physiological constraints. However, new insights into shallow and deep subsurface research may not necessitate harsh geochemical gradients or temperatures. Instead, restricted access to nutrients, vital for microbial activity, makes terrestrial subsurface studies of interest to investigate how moderately warm and/or acidic environments can still be uniquely constraining to microbiota.

In order to study subsurface microbiology, many continental deep subsurface observatories have been established (12, 17). Reaching depths of up to ~2.4 km, the Sanford Underground Research Laboratory in South Dakota has investigated the geochemistry and DNA sequences of up to 10,000-year-old groundwater to understand the energetics of chemolithotrophic organisms (15). These subsurface laboratories extend across multiple continents and countries, including the United States, Finland, France, Japan, and the United Kingdom (18). Another exists in Canada at the Kidd Creek Observatory, where fracture waters at 2.4 km in depth were acquired in order to cultivate microorganisms from the fluids and determine the metabolic energetic pathways from parallel geochemical analysis (19). Onstott et al. (17, 20) have established the presence of a microbial community in a gold mine in the deep subsurface from fluids obtained up to 3 km below the surface. These fluids proved microbiota indigenous to the host rock environment and further demonstrate the potential for life driven by equilibration of water-rock interactions. Microorganisms that exist in the secluded hematite iron formation of the Soudan Underground Mine State Park were examined with metagenomic analyses to determine genes representing diverse metabolic strategies from a low biomass system to overcome nutrient limitations (21).

These established observatories in the deep subsurface offer novel investigations into niche environmental conditions. However, the scope of which microorganisms survive in shallow subsurface environments from young meteoric groundwater infiltration from the surface versus chemical energy sources originating in local settings remains insufficiently answered (17). Studies have utilized deep fracture fluids or ancient aquifers to constrain life in the subsurface, yet only limited research has considered the influence of meteoric water penetration into the near subsurface. The role of young meteoric groundwater infiltration as a transportation mechanism for nutrients

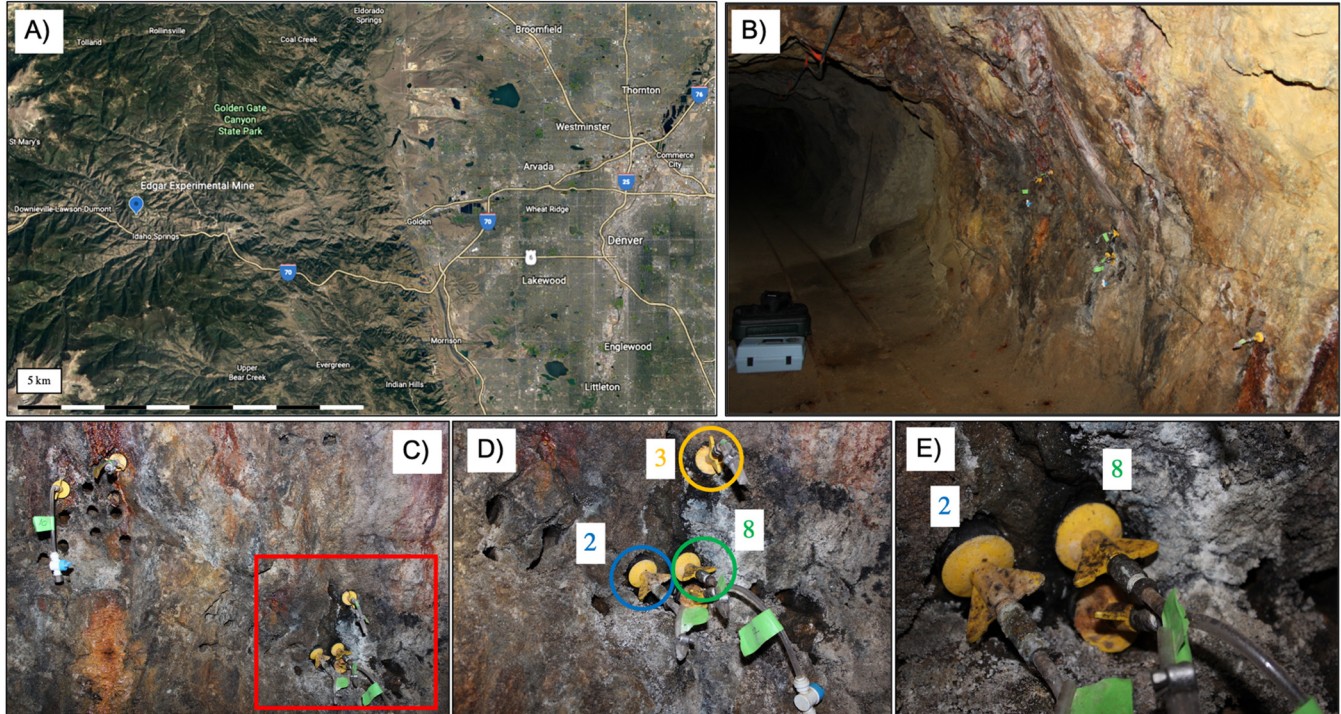

FIG 1 Composite images depicting the location of the Edgar Experimental Mine and the boreholes sampled within this study. (A) Landsat image of the location of Denver in reference to the mine in Idaho Springs following along interstate 70. (B) The sampling site for the study located in the area termed "C-Right" within the mine. Packed boreholes can be seen extruding out of the wall where suspected fluid leaks out and can be plugged up in order to collect fluid for subsequent sampling events. (C) Boreholes with packers included in the red box are boreholes observed in this study. Additional packed boreholes did not collect any fluid for sampling events. Previously drilled boreholes can be observed that do not leak any fluid potentially due to the complexity of the undefined fracture system. (D) A closeup image of the red box indicated from the previous figure with boreholes 2, 3, and 8 observed in this study. Borehole 6 (not captured) is approximately 5 m away on another wall of the field site. (E) A closer image of boreholes 2 and 8 located centimeters apart from each other; however, each collects distinctly different amounts of fluid at various rates of recharge.

introduces questions regarding the greater potential for life, the connectivity of the surface to the subsurface, and the relation of surface processes to subsurface microorganisms.

An additional consideration for deep observatories is the cost and accessibility of boreholes for sampling and inquiry. Due to expenses from boreholes and gaining access to subsurface environments, numerous questions remain unanswered from both marine and terrestrial subsurface ecosystem exploration (7). Therefore, easier access into the subsurface is advantageous not only to establish consistent research regarding the microbial communities existing in the subsurface but also to monitor geochemical variables influencing subsurface habitability with greater control over experimental design and statistical power.

This paper discusses the advantages of studying subsurface microbiology at the Edgar Experimental Mine, an easily accessible portal to the subsurface with ample opportunities to unveil and experiment with the subsurface biosphere. Once an active economic mining site, it was repurposed for teaching demonstrations by Colorado School of Mines for mining engineering practices. Four boreholes drilled from previous mining teaching practices are the focus of this study, located centimeters to meters apart along the same wall of our field site. Meteoric groundwater infiltrating the fractures of the mine leak out of the boreholes, offering the opportunity to study the subsurface microbiology of meteoric water influenced by young fluid-rock equilibration (Fig. 1). Additionally, the periodic (~2 weeks) to seasonal recharge of water makes it possible to study temporal influences through incubation and recharge periods. Here, we apply genomic, geochemical, and isotopic approaches to better define the variable microbial composition of borehole fluids and their relationship to surface influences from meteoric penetration into the subsurface.

## RESULTS

**Borehole geochemistry.** Over the 4-year sample collection period, the subsurface boreholes remained relatively uniform in their recorded analyte and physical chemistry measurements. While boreholes minorly fluctuate in analyte (major anion and cation) concentration, no discrete patterns were observed. Despite a lack of change, each borehole hosts nuanced geochemical profiles, providing niche microenvironments for microbial communities. Fluid recharge in the boreholes was unpredictable regarding volume and temporal repetition. Borehole 2 was able to consistently fill with greater than 1 liter of water each sampling period, whereas all other boreholes required months or years before refilling with fluid (Fig. 1). The remaining boreholes did not refill with predictable volumes of water at each sampling period, further emphasizing a complex fracture pathway for fluid transport. Physical parameters and chemical composition remained relatively homogenous among boreholes. Water temperature ranges varied seasonally, generally fluctuating between 10.6 and 13.5°C and reaching 17.6°C in summer. Dissolved oxygen varied most strongly from 1.97 to 7.7 mg/liter, while pH was generally uniform within individual boreholes but varied among all samples ranging from pH 5.30 to 7.16.

Ion chromatography (IC) and inductively coupled plasma atomic emission spectroscopy (ICP-AES) results of borehole fluids demonstrated weakly variable subsurface fluid chemistry. Concentrations of anions (putative electron acceptors for microbial growth as well as phosphorous sourcing) tend to be very low or undetected, allowing for selective substrate availability of microbial communities. Fluoride concentrations across all boreholes ranged from below detection or from 0.15 to 0.33 ppm. Nitrite concentrations were undetected except for three samples that recorded values of 0.19, 0.27, and 0.37 ppm. Nitrate remained undetected or reported values ranging from 0.45 to 1.27 ppm. Chloride concentrations varied unexpectedly within individual boreholes with concentrations between 1.59 and 22.08 ppm but spiked as high as 474.85 ppm. Sulfate was the most concentrated analyte among all boreholes and consistently remained high (usually greater than 1,000 ppm). Analytes chosen from IC and ICP analysis were selected to demonstrate greatest variability among the borehole fluids (Table S1 in the supplemental material). These analytes were evaluated with a principal-component analysis (PCA) to observe the relationship of geochemical measurements collected from the individual boreholes (Fig. 2). Of interest, principal component 2 appears to suggest a strong delineation driven by metals concentrations. In particular, manganese and zinc were the highest notable metal concentrations with maximum values of 27.15 mg/liter and 18.46 mg/liter, respectively. There is a linear relationship between the dissolved manganese and zinc concentrations in the borehole fluids (Fig. 3). This highlights the distinction between the metal-rich fluids found in boreholes 2 and 3 and more dilute fluids demonstrated in boreholes 6 and 8. Additionally, this relationship between the boreholes shows the consistency of borehole fluid chemistry despite sampling period and how this can serve as an indicator for variation in microbial community composition.

The $\delta^{18}$O and $\delta^2$H signatures for borehole fluids were compared to the global meteoric water line (GMWL) along with samples from other subsurface systems or observatories (Fig. 4). This trendline serves as an investigative tool for understanding if the subsurface fluids have undergone some extent of low-temperature water-rock interactions or remain isotopically conserved to reflect meteoric recharge from the surface (17, 19). Samples are plotted above the GMWL with slightly enriched $\delta^2$H values. Edgar Mine is located at greater elevation and drier climate, which often causes enriched $\delta^2$H values of meteoric water as a result from the rainout effect (22).

**Microbial community analysis.** 16S rRNA gene sequence analyses were performed on filtered water to determine the microbial community composition within each borehole. Several samples from each borehole were taken to contrast composition over time. The fluid samples were dominated by *Bacteria* (98.5%) with only a small presence of *Archaea* (1.5%) belonging predominantly to samples from borehole 2. All

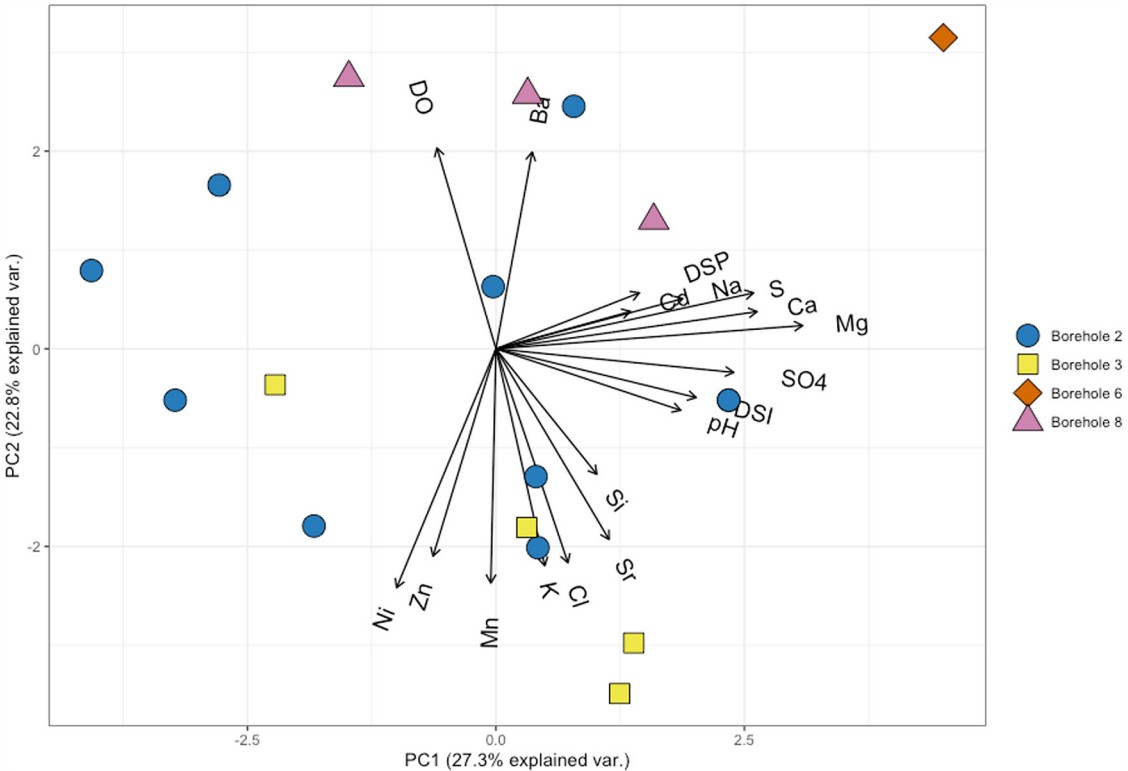

**FIG 2** Principal-component analysis (PCA) of geochemical data. Data are colored and shaped as the borehole the samples were collected from. The data have been centered and scaled to unit variance; DSP, days since previous extraction; DSI, days since initial extraction; var, variance.

boreholes were collectively dominated by the phylum *Proteobacteria* (39.2%) and also contained sequences for divisions of *Firmicutes* (16.1%) and *Bacteroidota* (13.0%) known to represent a diverse group of microorganisms.

Ordination via principal-coordinate analysis (PCoA) with a weighted UniFrac distance matrix was performed to determine statistical similarity of microbial communities from each borehole (Fig. 5) (23). Ordination suggested strong clustering as a result of grouping by borehole location. Additional soil samples were collected from the surface of the mine to confirm the distinctive compositions of borehole microbiota. An Adonis test was run on the dissimilarity matrix to verify sampling source as distinct and significant ($R^2 = 0.44$, $P = 0.001$). Weighted UniFrac distance matrices were also tested by Adonis for the number of days since the initial extraction of fluid from the boreholes, the number of days between sampling events, and the season in which samples were collected. Results demonstrated the number of days since the initial extraction period as significant ($R^2 = 0.38$, $P = 0.001$) as well as seasonality ($R^2 = 0.36$, $P = 0.001$) and the number of days between sampling events as modestly significant ($R^2 = 0.073$, $P = 0.047$). These values suggest that each borehole is host to unique microbial community compositions despite their relatively close proximity and similarity of host rock. The inherent covariance in our study between days since initial extraction and seasonality suggests that further work is needed to decipher these two signals. Nonetheless, a clear shift in community composition occurs over time, predominantly in boreholes 2 and 3. However, the source of this change requires additional sampling to account for unsampled seasons and how microbial diversity may change with even more time allowed to pass within the boreholes.

To further investigate the nuances in microbial community composition and shifts occurring over the course of the experiment, differential abundance tests revealed the top 15 most abundant genera in the boreholes and soil samples (Fig. 6 and 7). Boreholes 2 and

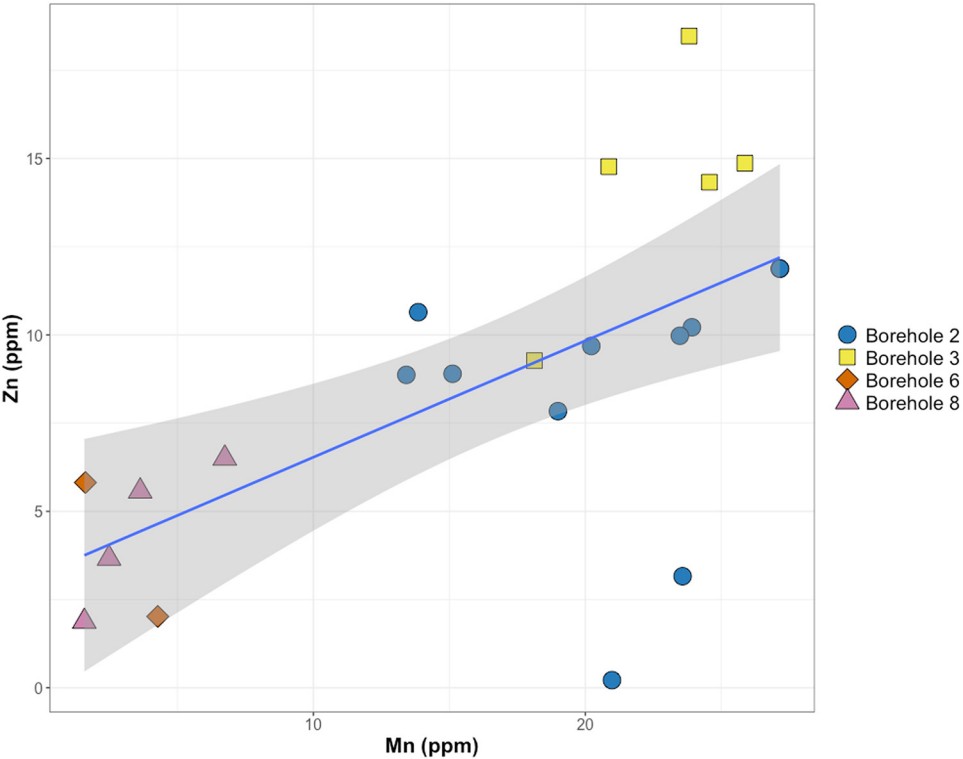

**FIG 3** The calculated linear regression of ICP-AES data for Mn and Zn concentration within the borehole fluids. The linear regression line is plotted in blue with 95% confidence intervals shaded in gray (adjusted $R^2$ = 0.3643, $F$ statistic = 13.04, $P$ = 0.00175). Samples are color coded from the borehole in which they were collected from. The figure also denotes the conserved relationship of metals concentrations within individual boreholes, despite the sampling interval.

3 appear to host similar microbial communities with subtle changes in taxa present and changes over the course of the sampling period. The boreholes share a large relative abundance of *Desulfosporosinus*, a potential sulfate-reducing bacteria, and *Sulfurifustis*, a potential sulfur-oxidizing bacteria, from each sampling occurrence (24–26). Boreholes 2 and 3 exhibit the same relationship where an increased relative abundance of *Desulfosporosinus* leads to a dramatic decrease in the relative abundance of other microbial members within the top 15 most abundant genera of fluid samples from boreholes 2 and 3. This stark contrast in microbial composition is demonstrated from the last two sampling periods, after 948 days have passed into the experiment (Fig. 6). In borehole 2, this shift appears to drastically change the microbial composition where the relative abundance of *Desulfosporosinus* continues to increase, and there is little relative abundance represented from additional microorganisms. In contrast, borehole 3 displays a shift in composition at this same sampling instance, but *Desulfosporosinus* decreases in relative abundance while other taxa increase (Fig. 6). Additionally, this shift does not last through the final sampling period, and *Desulfosporosinus* increases in relative abundance similar to earlier fluid samples. The other taxa present in the borehole fluid samples are indicators of metal-rich environments, including genera such as *Candidatus Nitrotoga* and *Chelatococcus* (26, 27).

Comparatively, the microbial community composition of boreholes 6 and 8 appear to be fixed without any distinctive influence as observed in boreholes 2 and 3. However, the genera observed in boreholes 6 and 8 juxtapose the microbial communities observed in boreholes 2 and 3, where the microorganisms represented do not indicate metal-rich environments. Instead, the microbial members from the fluid samples of boreholes 6 and 8 are typified by soil environments (28–32). Boreholes 6 and 8 do not display a clear temporal influence in microbial community composition and relative abundances. The lack of temporal evaluation of these two boreholes is likely owed to the unpredictability, and likely seasonality, of fluid recharge. Borehole 6

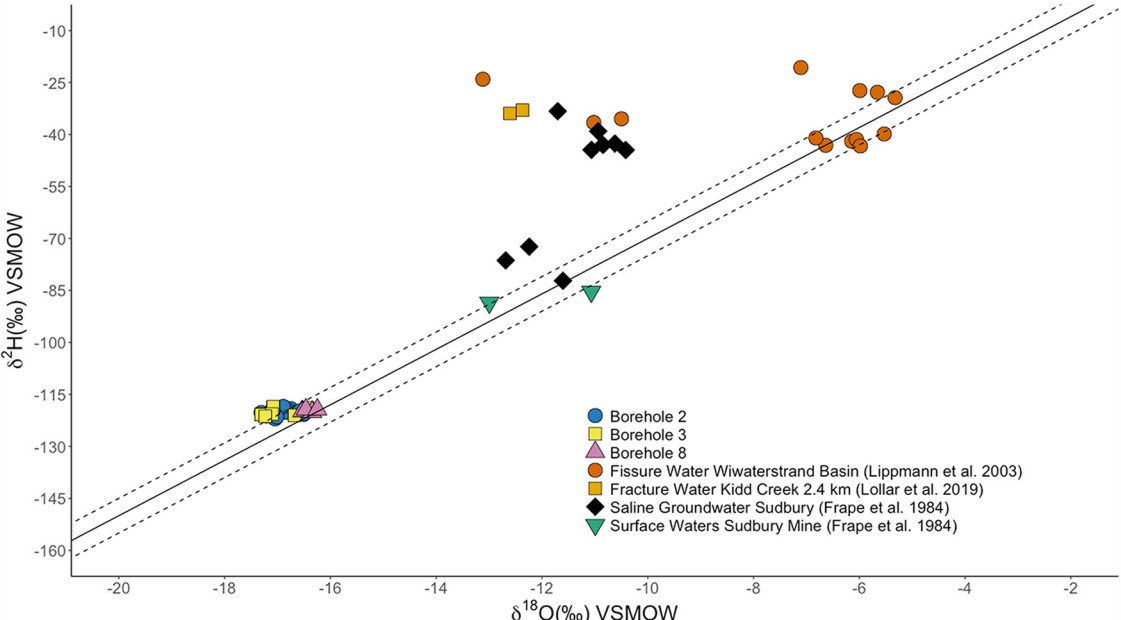

**FIG 4** The $\delta^{18}O$ and $\delta^2H$ signatures for borehole fluids collected at Edgar Mine. The solid line indicates the trendline for the global meteoric water line (GMWL) ($\delta^2H = 8 \times \delta^{18}O + 10$), while the dashed lines represent $\pm 5\permil$ trendline amendments. There appears to be no flux in isotopic values from each borehole, indicating that borehole fluids are likely a meteoric groundwater origin slightly enriched in $\delta^2H$, likely due to the rainout effect as a result of the location of Edgar Mine. These data demonstrate that there are no adverse processes altering the fluid isotope chemistry nor is there an alternate source for the fluids percolating throughout the boreholes. Additional $\delta^{18}O$ and $\delta^2H$ values are shown from the Witwatersrand Basin Mine (69), Kidd Creek Mine (19), as well as subsurface groundwaters and surface waters from the Sudbury Mine (70). VSMOW, Vienna Standard Mean Ocean Water.

contains few but prominent taxa, including the genera *Afipia* and *Sediminibacterium*. Borehole 8 fluid samples are represented by additional members such as *Aminobacter* and *Candidatus Nitrosotenuis* (Fig. 7). The only shared genus between the soil samples is *Bradyrhizobium*, which only appears in greater relative abundance from the latest sampling periods from borehole 8 fluids.

## DISCUSSION

The near-surface subsurface biosphere links the roles of surface and hydrogeological interactions driving habitability for microorganisms in subsurface environments. We have presented this biological investigation at Edgar Mine, a near-surface experimental/demonstration mine owned by the Colorado School of Mines, to establish the conditions influencing microbiology within the shallow subsurface. This study presents a process for determining the microbiology of subsurface borehole fluids from an active mine, contributing to understanding surface and near-subsurface interaction dynamics. By including the temporal facet of sample collection, we were able to evaluate the evolution of borehole fluids overtime and how successive recharge events affect microbial community composition.

**Geochemical dynamics.** The chemistry of the subsurface fluids in the Edgar Mine reflects the hydrologic context as well as the extent of equilibration with the host rock. Due to the quick recharging of borehole 2, it appears that residence time of meteoric water infiltrating the subsurface is quick. No significant relationship between the amount of fluid collected and chemical composition of each borehole is apparent. Chemical differences between boreholes are minor, which suggests small scale variations in fluid flow paths defined by a convoluted fracture system within the host rock, at least in this near-subsurface environment. Concentrations of metals are likely the result of fluid-rock equilibration that may be unique to each borehole's mineral inclusions. By extrapolation, this then has relevance whereby particular drill holes, mines sites, or other global subsurface access points are really detecting and capturing subsurface microbiology at precise screenshots within the subsurface with greater

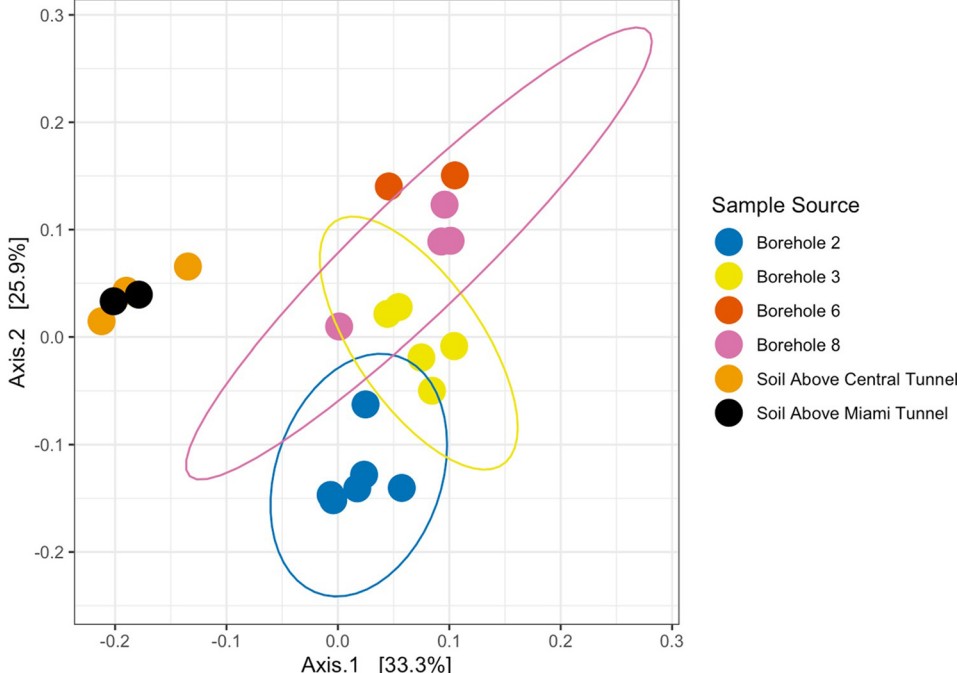

**FIG 5** Principal-coordinate analysis (PCoA) of weighted UniFrac distances between samples collected from borehole fluids and soil samples collected above Edgar Mine. Samples are color coded from the source they were collected from. Normal data ellipses are shown for data groupings when possible. Differences between the samples collected from each borehole and the soil samples are significant as verified by an Adonis test ($R^2 = 0.45$, $P \leq 0.001$).

variability being present than is actually determined by these single slices of data. The fracture system in the Edgar Mine remains to be fully investigated but requires more effort to decipher the link of surface influences into the subsurface and interconnectedness between the boreholes. In other subsurface geobiological data sets (both historical and future), caution should be taken in the application of "pin-prick" biopsy sample observations to the whole of the below ground.

$\delta^{18}$O and $\delta^{2}$H values plotted along the GMWL support a meteoric groundwater origin for the boreholes. Without a source of water from the deep subsurface or ancient aquifer, Edgar Mine poses as a unique boundary at the surface and near-subsurface interface. The isotopic values represented by the borehole fluids suggest that the water reflects the initial conditions of recharge from the surface. Further, this indicates that the water has a young geologic past that has not met the effects of low temperature water-rock interactions to cause $\delta^{18}$O and $\delta^{2}$H deviations from the GMWL (17, 19). However, the groundwater has been exposed long enough in the near-subsurface to equilibrate with the host rock, thus serving as an equilibrated nutrient source for host rock-associated microbiota. Meteoric groundwater percolated from the surface allows initial investigation into how the subsurface interacts with surface-derived fluids and how said fluid and boreholes evolve over time. The slightly enriched $\delta^{18}$O and $\delta^{2}$H values hold true to Edgar Mine's inland location at elevation and drier climate. Minor shifts in $\delta^{18}$O and $\delta^{2}$H values reflect seasonality but do not suggest any alteration of the fluid through supplementary subsurface processes (33).

The chemistry of Edgar Mine waters was further analyzed to understand biologically relevant chemical concentrations of borehole fluids and their effect on the composition of borehole microbial communities. Geochemistry of the borehole fluids varied weakly over the course of the experiment, which contrasts with heterogeneity in the composition of sequenced microbial communities. Instead, borehole geochemistry remained relatively consistent, displaying neutral conditions with low concentrations of analytes for potential microbial metabolic functions. Additionally, the incubation of fluid between

| | Borehole 2 | | | | | Borehole 3 | | | | |
|---|---|---|---|---|---|---|---|---|---|---|
| Firmicutes; Desulfosporosinus | 0 | 0 | 0.1 | 26.3 | 84.5 | 39.1 | 60.4 | 57 | 13.2 | 48.3 |
| Proteobacteria; Sulfurifustis | 17.1 | 8.9 | 15.6 | 15.7 | 0 | 0 | 0.4 | 2.4 | 6.5 | 1.3 |
| Proteobacteria; Candidatus_Nitrotoga | 7.8 | 23.6 | 24.4 | 1.7 | 0 | 0.1 | 0.3 | 0 | 0.6 | 0 |
| Bacteroidota; Sediminibacterium | 6.1 | 7.5 | 4.1 | 5.5 | 0 | 1.5 | 3.2 | 9.5 | 15.4 | 10.7 |
| Verrucomicrobiota; Candidatus_Omnitrophus | 11.4 | 8.7 | 12.3 | 12.1 | 0 | 0 | 0.2 | 1.3 | 5.4 | 0.5 |
| Actinobacteriota; CL500-29_marine_group | 13.2 | 10.6 | 7 | 5.9 | 12.6 | 0 | 0.4 | 1.5 | 3.6 | 0 |
| Actinobacteriota; Candidatus_Planktophila | 6.9 | 12.6 | 10.5 | 3.8 | 0 | 0 | 0.2 | 1.9 | 5.6 | 0.9 |
| Proteobacteria; Polaromonas | 4.7 | 0.5 | 0.2 | 0 | 0 | 18.2 | 5.6 | 2.6 | 5.8 | 8 |
| Proteobacteria; Pseudolabrys | 2.8 | 3.3 | 6.3 | 4 | 0 | 0.1 | 0.3 | 0.2 | 0.8 | 0.5 |
| Proteobacteria; Caulobacter | 1.6 | 0.4 | 0.3 | 0.1 | 0 | 4.4 | 6.2 | 2.4 | 4.9 | 2.2 |
| Bacteroidota; BSV13 | 0.4 | 0.8 | 0.8 | 3.1 | 0 | 2 | 1.7 | 3 | 6 | 2.7 |
| Proteobacteria; Chelatococcus | 0 | 0 | 0 | 0 | 0 | 3 | 2.2 | 4.8 | 6.6 | 4.5 |
| Proteobacteria; Sideroxydans | 5.1 | 2.3 | 4.1 | 2.4 | 0 | 0 | 0 | 0.3 | 1.3 | 0.5 |
| Proteobacteria; Afipia | 0.5 | 0.7 | 0.2 | 0 | 0 | 4.2 | 3.5 | 3 | 2.2 | 3 |
| Proteobacteria; GOUTA6 | 1 | 2.8 | 3.2 | 2.4 | 0 | 0 | 0 | 0.6 | 2 | 0.6 |
| Remaining taxa (116) | 21.4 | 17.3 | 11 | 17 | 2.8 | 27.5 | 15.5 | 9.5 | 20.2 | 16.3 |
| | 2017-10-24 | 2018-10-10 | 2018-11-16 | 2019-07-08 | 2020-06-24 | 2018-10-10 | 2018-11-16 | 2019-06-21 | 2019-07-25 | 2020-06-24 |

**FIG 6** The top 15 genus-level microorganisms within boreholes 2 and 3 as a function of time over the course of the experiment. Each column indicates the sampling date in which a sample for the respective borehole was collected. Numeric values within the box indicate the percent read abundance and is also represented as a heatmap through boxes colored by the numerical ranges represented from high (orange) to low (blue). The full phylogenetic lineage (phylum; genus) is depicted to give full coverage of the organism in abundance within each borehole.

extraction periods does not seem to alter fluid geochemistry. Solute transport may not play as significant a role between the surface and near-subsurface influencing borehole fluids given the lack of geochemistry that would reflect soil origination, although additional analyses of dissolved organic carbon and nitrogen would aid in this inquiry. Nonetheless, in the more nutrient-limited deep subsurface, the movement of dilute geochemical analytes from above to below ground is likely critical to microbial metabolism.

Inorganic nitrogen concentration was low or below detection, suggesting the near-subsurface boreholes of the Edgar Mine could be a nitrogen-limited system. Similarly, phosphate concentrations were below the detection limit, and general phosphorus concentrations were extremely low, further suggesting a phosphorus-limited environment. One key factor driving potential microbial metabolism and influencing fluid geochemistry is the elevated sulfate concentrations. Sulfate is highly concentrated in the borehole fluids and can pose as a major component in sulfur-reducing metabolism (34, 35). The positive correlation between manganese and zinc (Fig. 2) demonstrates a relationship between these metal concentrations and where the borehole fluid source was collected, which can be indicative of the host environment for microbial communities (36–38). Similarly, the linear fit observed between the manganese and zinc concentrations reflects the conserved relationship between the two metals within individual boreholes. Boreholes 2 and 3 contain the greatest amounts of metals, while boreholes 6 and 8 consistently contain the lowest concentrations despite the sampling period. This gradient distinguishes the metal-rich fluids versus the more dilute fluids that can host diverging metabolic potentials.

**FIG 7** The top 15 genus-level microorganisms within boreholes 6 and 8 as a function time over the course of the experiment and soil samples collected above Edgar Experimental Mine. Each column indicates the sampling date in which a sample for the respective borehole was collected. Numeric values within the box indicate the percent read abundance, also represented as a heatmap through boxes colored by the numerical ranges represented from high (orange) to low (blue). The full phylogenetic lineage (phylum; genus) is depicted to give full coverage of the organism in abundance within each borehole or soil sample.

**Borehole microbial community analysis.** Groundwater collected from packer install-ment allowed for fluid-rock equilibration spanning weeks to months between subsequent sampling events. While the undefined fracture system produces an unpredictable volume of fluid, both the rock and mineral composition appears to be unique to individual bore-holes at the centimeter scale. The chemical gradient established within the more metal-rich boreholes is associated with a shift in community composition, juxtaposed by more dilute boreholes. All boreholes are filled with similarly sourced meteoric water infiltrating the subsurface, yet each borehole displays a unique community composition as deter-mined from PCoA weighted UniFrac analysis (Fig. 5) and microbiota relative abundance investigations (Fig. 6 and 7). Phylogenetic relatedness determined by PCoA analysis of borehole microbial communities suggests that niche microcosms of microorganisms can establish themselves in boreholes located only centimeters to meters apart. This finding indicates that subsurface microbiology may be more sensitive to fluid-rock equilibrated systems at a finer spatial resolution than previously thought or understood. The host rock composition allowing for the dissolution of more metals or anions into fluids may be a key driver for supporting microbial habitability as well as developing an aerobic/anaerobic environment. While dissolved inorganic carbon (DIC) and dissolved organic carbon (DOC) compounds were not investigated, carbon sources within each borehole likely influence microbial community composition. A model considering all carbon and electron donor and acceptor modulations that drive differences in our samples would be speculatory; nonetheless, subtle fluid variations in DOC and dissolved organic nitrogen (DON) likely assist in structuring the microbial heterogeneity that is unexplained by measured anions and cations alone.

There appears to be an evolution of microbial community composition as the experiment progressed, reaching greater than 800 days. Previous studies have deter-mined the effect of pre- and postpacker installment in establishing changes in micro-bial composition (39). The installment of a packer into a borehole allows for fluid to equilibrate with the host rock and provides an opportunity for metals and other nutrients to become available to microbial communities. Similar studies have investi-gated boreholes of deep mine systems, yet not on such a frequent and lasting

temporal scale or within a near-surface environment (17, 19). The Edgar Mine boreholes are supplied by frequent seasonal recharge events, which may vary in geochemical composition when transported through the subsurface. This nonconsistent recharge could play a role in determining the habitability of the near-surface subsurface biosphere.

Intriguingly, the time between sampling periods did not demonstrate a significant correlation to microbial community composition (Fig. S1 in the supplemental material). This finding suggests that defined time intervals between borehole fluid extractions do not dictate the composition of the microbial community within a borehole. Instead, the temporal influence is sourced from moderate changes over a greater period of time, reflecting stages of evolution of the microbial community in borehole fluids. This finding highlights that microbial communities found in the shallow subsurface borehole fluids are not predictable based on the time between extractions. The season in which samples were collected also appears to display a strong correlation. This seasonal signal is difficult to discern and is biased from overlapping signal produced by the days since the initial fluid extraction. Nevertheless, seasonality may introduce an interesting avenue of further examination to determine the source of groundwater and how this may alter the near-subsurface microbial community (40, 41).

Snowfall may carry large amounts of biodiversity, especially through storm events, which could introduce nonindigenous microorganisms into the subsurface (42). Transportation of microbial members through snowmelt has been speculated at the Henderson Mine near Empire, CO, from the presence of *Chloroflexi* phylotypes (39). The presence of *Polaromonas* in boreholes 3 and 8 offers the same insight given the ≥98% identity match through BLAST search of sequences for a species isolated from an ice core (43). The role of snowmelt transportation into the near subsurface requires further investigation to elucidate the role of meltwater and other seasonal hydrological infiltration into the subsurface as a mode for microbial transportation. Additionally, the role of water extraction on overlying litter/soil communities with subsequent seepage into the subsurface could further explain source and seasonal variation in near-surface subsurface environments. The role of hydrology intercepting the subsurface and introducing microbial diversity has direct relevance toward agriculture and home use that rely on near-surface water wells. Hydrological transport of microbial communities is also tied to the ever-present effects of climate change, which—through the mechanism of ground infiltration—will inevitably play a role in altering the geochemical and microbial composition of the subsurface, a concept likely also relevant to natural caves on Earth.

Geochemical gradients determined by metal concentrations (by manganese and zinc) reflect the presence or absence of potential endemic subsurface microbial communities between borehole fluid samples. Boreholes 6 and 8 culminate an assembly of microorganisms typified by nonsubsurface processes (Fig. 7). It should be noted that many of these organisms require aerobic conditions that could have been established from installment of the packer system (31, 39). When considering the microbial communities of these borehole fluids from the Edgar Mine, there is potential for contamination from operations within the mine. Specifically, the role of human activity, drilling fluids, and machinery used to drill the boreholes and install the packers in this demonstration and teaching mine cannot be ignored. This can make particular members of the microbial community difficult to determine if they are indigenous to the subsurface borehole ecosystems or a result of mining operations (44). Nonetheless and despite this caveat, the composition of the microbial communities observed in boreholes 6 and 8 may suggest a surface-influenced subsurface community. In particular, these two boreholes demonstrate the only quantifiable concentrations of nitrogen in the form of nitrate (Table S1). The presence of soil-originating microbes with known nitrogen-related metabolisms, such as nitrification displayed by *Candidatus Nitrosotenuis* may suggest that any residual nitrogen compounds are quickly expended and are not left dissolved as inorganic fractions in the borehole fluid (32). The presence of *Bradyrhizobium* suggests the possible infiltration of soil-derived microbiota, as it is a known nitrogen-fixing bacteria commonly found in plant soils (43). *Bradyrhizobium* is

the only shared genus between these two boreholes and the soil samples. This makes it difficult to assess the true transport mechanism of soil-derived microorganisms that are being introduced into the subsurface versus contamination from mining operations and installation of borehole packers. Additional members such as *Sphingobium* make it clear that the microbial communities observed in boreholes 6 and 8 do not originate in the subsurface given the known metabolisms of phylotypes identified in the fluids (30). Boreholes 6 and 8 contain microbial communities that suggest they were transported into the near subsurface via a mechanism outside an endemic subsurface environment. The lack of overlap between shared relative abundance of microbial composition between boreholes 6 and 8 and the soils confounds how nontraditional subsurface microorganisms penetrate the subsurface. Thus, the interaction between the surface and subsurface requires greater scrutiny in order to resolve how subsurface microbial communities can establish through surface infiltration or mining practices. The development of surface-based communities in the subsurface remains to be fully explored yet should press warning into determining true endemic members of the subsurface in any subsurface location. The proximity of boreholes 6 and 8 to boreholes 2 and 3 illustrates how the spatial resolution for subsurface microbiology may be finer than we understand. The need for greater data collection emphasizes the potential of undersampling over both spatial and temporal scales and reflects our lack of knowledge toward the true diversity and dynamic capabilities of microbial ecosystems in subsurface investigations.

The distinct differences in microbial community composition between boreholes is highlighted in the PCoA weighted UniFrac analysis. Boreholes 2 and 3 contain microbial members that suggest the presence of a more native community of microorganisms to the boreholes of the Edgar Mine. The presence of the family *Gallionellaceae* reflects the greater concentrations of metals in the fluids of borehole 2 given the known iron-oxidizing and potential manganese-oxidizing metabolic capabilities (45, 46). The inclusion of additional members associated with metabolic potentials involving metals further supports microbial members that have adapted to metal-rich fluids. Members of the family *SR-FBR-L83* are observed to respire electron acceptors, such as Fe(III) and nitrite (26) (Fig. S2). Similarly, the genera *Chelatococcus* requires metal-chelating compounds to use as sole energy sources (27). Meanwhile, *Candidatus Nitrotoga* is capable of adapting to nitrite-limited conditions and perform sulfur-oxidizing metabolic functions (47). Moreover, the presence of genera *Sulfurifustis* and *Desulfosporosinus* indicate metabolic potentials for sulfur oxidation and sulfate reduction, respectively (24, 25). The prominence of sulfate would suggest that sulfur metabolisms play a strong role; however, lack of knowledge of dissolved organic or inorganic carbon compounds makes it difficult to decipher the metabolic pathways occurring in these boreholes (34, 35).

Inclusions of sulfide veins would offer direct microbe-mineral interactions to support sulfur-oxidizing metabolisms (48). It is difficult to discern the bioenergetics observed in borehole 2, but, given the low concentration of dissolved oxygen with available electron acceptors (e.g., sulfate and iron), it is feasible that chemolithotrophy is a prominent metabolic process occurring in the metal-rich boreholes (49). The correlation of metal-rich fluids observed in boreholes 2 and 3 to host distinct microbial communities is supported by our identification of the most abundant taxa that are not present in boreholes 6 and 8 or in the overlying soils. Further work is necessary to understand how metal-rich fluids can dictate the diversity of microbial communities and how to use this as a tool for recognizing hot spots of microbial activity.

Microbes respiring metals for electron donors and acceptors are incorporated in the fluid that has equilibrated with the host rock. The presence of these microbes in boreholes 2 and 3 offers speculation for preexisting biofilms that are equilibrating with the infiltrating groundwater (50, 51). Leaking water from unpacked boreholes has already produced visible discolorations along the walls of the mine (Fig. S3). Brown and yellow discoloration with white precipitates suggests a prospective environment for biofilms with rich mineralization to develop. Further investigation into preexisting biofilms could benefit from examining boreholes without leaking fluids. Installation of a packer

and injecting sterilized fluid into the borehole could determine how the host rock equilibrates with the fluid and what microbiology may be native to the borehole (52, 53).

Tools for microbiology to aid in economic mining practices have yet to be fully investigated. Our study demonstrates how slight nuances in borehole microbiology can occur at centimeter to meter scales, and these measurable differences occur at a greater rate than differences in fluid geochemistry. Understanding how microbial community composition reflects mineralogy and potential ore formations could be a useful tool to aid the development of less invasive mining practices. High sulfate concentrations are reported within all of our boreholes, which suggests potential bioleaching of low-grade ore and sulfide veins could be occurring, and this would not be unexpected in Colorado (48, 54, 55). Using molecular biological-based tools to investigate and identify ores or resources of interest would pose a more environmentally friendly and economically sustainable method of resource detection (48, 56). In conjunction, the ability to tie microbial diversity with predictive hydrological behaviors may be needed to demonstrate the connectivity of surface and subsurface interactions. Statistical models combined with molecular data could generate a powerful tool in understanding the path and composition of subsurface fluids.

The work we present here demonstrates the capability for the Edgar Mine to establish itself as a premier subsurface facility for microbiological experimentation. This is the first full molecular biological research experiment conducted at the Edgar Mine, an experimental teaching demonstration mine that was created to train people in mining engineering practices, and this study unveils the complexity of boreholes within a spatial distance of centimeters to meters apart. Our work also begins to investigate the role of the shallow subsurface and the dynamic relationship it shares with surface influences into the near subsurface. The ability to define how the subsurface biosphere interacts and is possibly shaped by surface infiltration still needs further investigation to contrast the deep biosphere. Understanding subsurface microbial diversity is an important tool for determining potential sources of mineralogy and ore deposits for less invasive mining practices and will also be essential in unveiling biogeochemical cycling of the subsurface at numerous locations around Earth as well as other planetary bodies.

## MATERIALS AND METHODS

**Field site.** Fluids from boreholes previously drilled in the Edgar Experimental Mine were collected beginning in fall of 2016 and extending through summer of 2020. The boreholes were created as a function of mining and drilling demonstrations for teaching purposes by the Colorado School of Mines, who also own the facility and guide operations. Edgar Mine contains thousands of available boreholes, and more continue to be drilled as a part of training efforts for students at the Colorado School of Mines interested in mining practices. Edgar Mine is located in Idaho Springs, CO (39°44′50.21′′N, 105°31′31.39′′W; 2,405 m), and acts as a repurposed gold, silver, lead, and copper mine for teaching and research initiatives incorporating mining engineering and safety. As a function of demonstration practices, boreholes were drilled into the walls of the site named "C-Right" (Fig. 1). Spatial resolution of the boreholes varies on the scale of centimeters to meters in distance of each other and differ greatly by orientation of drilling angle. The mine is composed of Precambrian rock, primarily by gneisses, where the specific field site of "C-Right" is dominated by quartz-plagioclase gneiss and quartz-plagioclase-biotite gneiss (57).

It was observed that many of the boreholes were leaking fluid, suggesting a complicated fracture system allowing for opportunistic boreholes to collect substantial amounts of fluid. This fluid serves the potential for many water-rock interactions fueling microbial habitability. Similarly, fluid buildup for extended periods of time can develop unique water-rock equilibration, hosting niche microenvironments in separate boreholes for diverse microbial communities (54, 55). In order to take advantage of the leaking fluid, boreholes were fitted with custom-built expandable packers at the mouth of the borehole to trap leaking fluid (Fig. S4 in the supplemental material). These fluids were then left to equilibrate with the rock and develop microbial communities on temporal scales of just over 2 weeks to over 1 year to assess temporal variability within sampled boreholes. The field site "C-Right" within the Mine is located approximately 500 feet below the surface. We propose that meteoric groundwater is the main source of fluid infiltration into the subsurface, following an undefined fracture system allowing certain boreholes to fill with greater amounts of fluid than others. Infiltration of surface fluids also possess potential to navigate toward the subsurface boreholes and influence both geochemical and microbial processes differentially over space and time.

**Sample collection.** Collection of fluid was accessed from the expandable packer at the mouth of each borehole to assess both geochemistry of the fluid and microbial community composition. A sterile syringe was used to draw fluid trapped behind the packer and passed through a 0.22-$\mu$m filter to collect biomass. Filters were immediately frozen on dry ice in the field and stored at $-20°C$ back in the lab until further

extraction and processing. Following initial extraction of borehole fluid, the preliminary 50 ml of fluid was used to record physical chemistry measurements of dissolved oxygen, pH, and temperature immediately using a daily calibrated Hach multiparameter field meter (HQ40D, Hach, Inc., Loveland, CO). An additional 50 ml and up to 1 liter of filtered fluid were collected in clean autoclave-sterilized glass vials for each borehole for downstream geochemical and water isotopic analyses. To contrast surface microbiology (a potential high-biomass origin of subsurface-infiltrating microbiota), soil samples were collected above the "C-Right" field site and put on dry ice in the field and stored at $-20°C$ until further processing. All further analysis and preparation were conducted in the laboratory. Samples were collected over various time scales in order to assess temporal variability of microbial community composition and geochemical fluctuation. Borehole fluid was allowed to incubate for as short as 17 days and as long as 351 days within certain boreholes. Due to the unpredictable nature of the fracture system, some boreholes collected greater quantities of infiltrated fluid than others and, thus, were subjected to more frequent sampling opportunities. Predetermined incubation periods (2 weeks) were used to assess how microbial community composition may alter from human-chosen time intervals. Similarly, the amount of time passed since the initial fluid extraction was also recorded to observe overall temporal effects shifting microbial diversity.

**Chemical analysis.** Filtrate was collected and analyzed for major ion concentrations through ion chromatography (IC) and inductively coupled plasma atomic emission spectroscopy (ICP-AES) spectroscopy. IC analyses were run on a Dionex ICS-900 IC system, while ICP-AES was performed on a PerkinElmer 8300 ICP-AES. ICP samples were acidified with 3 to 5 drops of 10% nitric acid with a total volume of 12 ml for sample submission. Additional fluid collected from boreholes was run on a Picarro water isotope analyzer (D/H and $^{18}O/^{16}O$) with cavity ring down spectroscopy (Picarro, Inc., Santa Clara, CA, USA) in order to determine source water variability of the borehole fluids.

**Molecular microbial analysis.** DNA from filters used to collect biomass from borehole fluids and soil samples was extracted with the ZymoBIOMICS DNA miniprep kit (Zymo Research, Irvine, CA, USA) according to the manufacturer's instructions. The concentration of raw DNA extract was quantified with the Qubit double-stranded DNA (dsDNA) high-sensitivity assay (Thermo Fisher Scientific, Waltham, MA, USA). PCR amplification of the V4-V5 region of 16S rRNA gene was performed with the 515-Y-M13 forward (5'-GTA AAA CGA CGG CCA GTC CGT GYC AGC MGC CGC GTT AA-3') and 926R reverse (5'-CCG YCA ATT YMT TTR AGT TT-3') primers. The reverse primer was illustrated by Parada et al. (58), and the forward primer contains the M13 forward primer sequence (indicated with underline) to aid in an additional PCR amplification for barcoding attachment (for subsequent DNA sequencing) along with the 16S rRNA gene-specific sequence (the remaining sequence not underlined), as performed in references 58 and 59. A second PCR amplification step was then included for the addition of barcodes to the M13 region of the forward primer as adapted from reference 60. Initial PCR was performed on a Techne-TC 5000 thermocycler with the following parameters: initial denaturation at 94°C for 2 min, followed by 30 cycles of 94°C for 45 s, 50°C for 45 s, and 68°C for 1 min and 30 s and ending with a final extension at 68°C for 5 min and a hold at 4°C until removal from the thermocycler. Barcoding of sequences was performed on a limited 6-cycle PCR on the purified initial PCR product. PCR product cleanup was performed with a $0.8\times$ concentration of KAPA Pure beads (KAPA Biosystems, Indianapolis, IN USA) according to the manufacturer's instructions. Sequencing was performed on an Illumina MiSeq at either the Duke Center for Genomic and Computational Biology (Raleigh, NC) or the University of Colorado, Denver, Anschutz Medical Campus. The resulting FASTQ files were demultiplexed and trimmed with AdapterRemoval2 (61). DADA2 was used to filter reads by error rates, amplicon sequence variants (ASVs) were identified, paired-end reads were merged to construct a sequence table, and chimeric sequences were removed (62). Taxonomic assignments were called with the SILVA small subunit (SSU) database training file (version 138). A phylogenetic tree was created with the "phangorn" package in R by first constructing a neighbor-joining tree then fitting a generalized time reversible with gamma rate variation maximum likelihood tree (63). ASV tables, phylogenetic trees, taxonomy tables, and sample metadata were collected into a phyloseq object for further downstream analysis (64). Lab blanks that were a part of the DNA extraction process were included in the sequencing run in order to identify potential contaminants within our borehole fluid and soil samples by manually eliminating ASVs identified in the lab blanks from our samples through a highly sensitive curation (65).

**Statistical analysis.** All microbial community and geochemical multivariate analyses were conducted and produced in R. A principal-component analysis (PCA) was compiled with each sample being ordinated to assess geochemical drivers of the system. The PCA was created using the base R "stats" package (66). To assess microbial communities of the boreholes, all sample reads were rarified to an even depth of 5,761 reads per sample to remove as few reads as possible while maintaining enough diversity among samples, with threshold determined by rarefaction curves (Fig. S5). The samples were also ordinated though principal-coordinate analysis (PCoA) with a weighted UniFrac distance measure in order to consider phylogenetic relatedness as well as relative abundances. All figures were generated and visualized with the "ggplot2" or "phlyosmith" package (66–68).

**Data availability.** The raw, unmerged 16S amplicon sequences as forward and reverse read files are publicly available on figshare at https://dx.doi.org/10.6084/m9.figshare.14740791. The code used for this study and additional metadata are available from our GitHub page at https://github.com/pthieringer/Edgar-Mine.

## SUPPLEMENTAL MATERIAL

Supplemental material is available online only.

**SUPPLEMENTAL FILE 1**, XLSX file, 0.02 MB.

**SUPPLEMENTAL FILE 2**, PDF file, 0.7 MB.

## ACKNOWLEDGMENTS

We thank Matt Schreiner, Lee Fronapfel, and Clinton Dattel for insightful collaboration with safety training and experimental design at the Edgar Experimental Mine. We are grateful to Katherine Dawson at Rutgers University for guidance and analysis of water isotope data.

This work is supported by National Science Foundation Graduate Research Fellowships (grant no. 1646713): fellow identification numbers 2018254777 (P.H.T.) and 2019258966 (A.S.H.). J.R.S. is supported by a NASA Exobiology grant, number 80NSSC19K0479. The funders had no role in study design, data collection and analysis, decision to submit for publication, or preparation of the manuscript.

We declare that the research herein as conducted in the absence of any commercial or financial relationships that could act as a potential conflict of interest.

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
