## [Reviewer comments · Microbiology Spectrum]

Microbiology Spectrum

Spatial and temporal constraints on the composition of microbial communities in subsurface boreholes of the Edgar Experimental Mine

Patrick Thieringer, Alexander Honeyman, and John Spear

Corresponding Author(s): John Spear, Colorado School of Mines

Review Timeline:

Submission Date:	June 28, 2021
Editorial Decision:	July 26, 2021
Revision Received:	October 5, 2021
Accepted:	October 7, 2021

Editor: Jeffrey Gralnick

Reviewer(s): The reviewers have opted to remain anonymous.

Transaction Report:

DOI: <https://doi.org/10.1128/Spectrum.00631-21>

July 26, 2021

Dr. John R. Spear
Colorado School of Mines
Environmental Science and Engineering
1301 19th Street
Golden, CO 80401

Re: Spectrum00631-21 (Spatial and temporal constraints on the composition of microbial communities in subsurface boreholes of the Edgar Experimental Mine)

Dear Dr. John R. Spear:

Thank you for submitting your manuscript to Microbiology Spectrum. Two reviewers with expertise in the field have provided extensive feedback on your manuscript. While both found the work interesting and were enthusiastic about the "ingenious" sampling method and topic matter, they also identified major issues with the language and data interpretation. While the comments below are extensive, they will no doubt lead to an improved manuscript which I very much look forward to seeing.

When submitting the revised version of your paper, please provide (1) point-by-point responses to the issues raised by the reviewers as file type "Response to Reviewers," not in your cover letter, and (2) a PDF file that indicates the changes from the original submission (by highlighting or underlining the changes) as file type "Marked Up Manuscript - For Review Only". Please use this link to submit your revised manuscript - we strongly recommend that you submit your paper within the next 60 days or reach out to me. Detailed information on submitting your revised paper are below.

Link Not Available

Sincerely,

Jeffrey Gralnick

Journals Department
Reviewer comments:

Reviewer #1 (Comments for the Author):

The manuscript "Spatial and temporal constraints on the composition of microbial communities in subsurface boreholes of the Edgar Experimental Mine" by Theieringer et al. seeks to quantify the microbial community composition within fluids emanating from experimental boreholes at the mine. Using mines to access subsurface microbes is an excellent and cost-effective way to investigate the deep biosphere. The authors use an ingenious, simple packing system to isolate the boreholes and to collect fluids. Overall, I thought the manuscript needs condensing, especially the introduction. Additionally, I'm concerned that the authors have not properly analyzed their dataset correctly. The methods are sparse and so it is difficult to assess what they have actually done with the amplicon reads. DADA2 has lots of nuances and caveats that can lead to screening of sequences and so I'm worried that one reason the number of reads/sample is so low, is that DADA2 is screening out real data. I've highlighted this down below. Furthermore, it seems that kit contamination may be a problem, due to low biomass of their samples. There are several papers describing this and one recent one for subsurface systems (see below). Please be aware that this can be a large

problem for subsurface systems. I think this is an interesting system that should be highlighted but your narrative needs focus to really bring that out.

L 59: "analogs to.." instead of analogous to...

L60-64: The focus on marine systems here seems out of place. I understand what you are going for but still seems out of place or maybe needs to be reworded a bit.

L 69: Fount seems to be the wrong word here. I take it to mean source but seems strange. Maybe biodiversity would work better?

L 80: I don't think the sources energy of chemolithoautotrophy are unique in hydrothermal settings. You still see the same redox couplings in terrestrial systems both in the deep subsurface, vadose zones, and surface. Maybe reword?

L91-95: This seems out of place and somewhat off topic as you introduce the paragraph discussing nutrients and then talk solely about water and not the nutrients that are dissolved.

L96: another laboratory to look at is Soudan Iron Mine, see Sheik 2021 Frontiers.

Introduction thoughts: I think the intro needs some tightening and reworking. All the ideas are there but could be presented more succinctly.

Results:

L154: when discussing ions avoid using levels. You are talking about the chemistry of the system and so use concentration instead.

L 149-165: While everyone should be accustomed to element's shorthand you should probably still use the full word, like Nitrate (NO₃⁻) or nitrite (NO₂⁻). Also this section could also be condensed a bit.

Figure 2/table2: You have a lot of Geochem data here in Table 2. Maybe put this all in supplemental and use a RDA style analysis to show how the boreholes relate to each other and which elements/anions are significantly delineating the boreholes. Its unclear why you would focus only on Zn vs Mn

L 166: This may be just my pet peeve but starting out a paragraph with "Figure 3" seems odd. I think rephrasing this sentence to frame what and why you are measuring would be better.

Figure 3: I think you need to plot some of the other subsurface values for some of the other mines you mention in the introduction. Place your values into context with other mines/study sites.

Microbial analysis

L 181-185: It is difficult/impossible to see any of this data since borehole 2 is only shown at the phylum level. You also have a lot of discussion happening in your results section. Either combine into a results/discussion or move to discussion

Figures 4/5/6: While this is a time series, using line graphs to depict this type of data is not great. I think showing us a stacked bar graph of all the samples would perhaps be more useful to see how changes are occurring relative to one another. Perhaps just expand figure 6 to include all the boreholes.

Observation: I think starting with the PCoA results would better set up your taxonomic analysis of the boreholes. Establishing there is a difference and then telling us what is driving the difference would be good.

Taxonomy: You are using a lot of handwaving in this section and combining discussion. If that is your intent change to a combined results/discussion. Did you do any statistical analysis like LefSE or random forest modelling to pull out taxa that are significantly different in the boreholes? Feels like that would be a better approach here.

You mention that these are all soil-related groups. Is there any direct sharing of sequences that are from the soil and found in the subsurface samples?

Discussion:

In your methods you don't mention that you screened your sequences from kit contaminants. Sheik et al 2018 show that for low biomass subsurface systems this is necessary and also list common genera that need to be screened and/or highly scrutinized. Your abundance of "soil genera" in your low-flow boreholes would suggest that kit contamination may be a large problem. I would definitely suggest being very careful with your taxonomic analysis.

You say that there is anaerobic chemolithoautotrophy several times. However, your O₂ in the waters is quite high in every borehole. If you have any reduced fluids that are interacting abiotic oxidation is going to be a problem, especially with sulfur and definitely with iron.

Methods:

In general, your methods are very sparse. For instance, to generate a Unifrac metric, you need a phylogenetic tree. How did you produce this? You say you used a modified pipeline and then cite a paper. Did you use their pipeline they developed for ASVs generation or did you modify theirs in some way? These are all very important details that need to be in the paper.

You're using two different places to sequence, did you send this off samples multiple times at each facility or was is all just 2 bulk runs? In either case you need to specify if you ran DADA2 independently on each run then combined all the runs together with DADA at the end to create an ASV table. This will definitely have downstream affects with the model prediction for the ASVs, since it is well documented that there are inconsistencies between sequencing facilities.

Why did you get such low recovery of ASVs? With MiSeq you should be getting well over 15K Seqs per sample. Are these being screened out during the ASV creation? The chimera removal step from Dada2 is very sensitive and can screen out real sequences. Did you check to see if this is happening? See above for some reasons why. But also how did you remove your primers/adapters?

L 542: I think the name of the Duke Center you used is incomplete or is mistyped

Figures: Be cognizant of what your figures are trying to convey. You have lots of data but the figures don't necessarily show this. Please use different shapes for each borehole and use colors that are color-blind appropriate. Several of the colors bleed into each other making it tough to discern which borehole/location they belong to.

Reviewer #2 (Comments for the Author):

Summary: This manuscript reports on a long-term study of the geochemistry and microbiology of subsurface fluids from boreholes within the Edgar Experimental Mine. The authors observe differences in geochemistry between sites, and concurrent changes in microbial community composition, even at close spatial scales. While the work itself is interesting, the logical framing of key aspects of the work, the scholarship in interpretation of both shallow subsurface geochemist and microbial physiology, and the writing itself all present major problems which should preclude publication at this time.

Major Comments:

I commend the authors on completing this long-term field study of subsurface microbiology, a rare feat. The study itself is interesting and I think with sufficient rewriting should be publishable. That said, the way this is written and presented is currently ineffective and will require major revisions.

I refrained from line editing this document, but it needs a thorough polish and cut from a senior editor. Most of the writing is verbose and includes many clauses where meaning is not clear and subject verb disagreement precludes precise interpretation. There are many superlatives and colloquialisms which interfere with clearly conveying the scientific messages. There are also structural problems, namely, a lack of necessary information in the introduction, lots of interpretation is present in the results, and the discussion is circuitous and repetitive (often repeating parts of the results). When rewriting please try to be more precise with the presentation of the study data and more thorough in referencing information needed for interpretation.

Scientifically there are also some major problems.

The interpretation of fluid origin in contrast to microbial origin is problematic. Most groundwater (even very old ground water) is of meteoric origin, but that doesn't mean it comes in directly from the surface. A groundwater origin is discounted in favor of a meteoric origin (which again is not mutually exclusive), but this interpretation of extremely young fluids is not well supported. It is also not consistent with the surface connectivity or recharge times, or fluid chemistry reported for each borehole and reflected in the microbial diversity data.

Interpretation of physiology from tag sequencing data is always a risky business and must be done with considerable care. To interpret physiology, you must have a very close match with cultured organisms, not just at the family level, but by evaluating your individual ASVs, stating how close they are to organisms with physiology information, and then proceeding with cautious interpretation. Also, looking at relative abundance data means that when one thing goes down, others go up, but that does not mean that the actual abundance of those organisms went up. There is no mention of diversity metrics or other normally ecology statistics which might help get around some of these challenges. I think the display of taxonomic data could be far better as well. I think the heat map shown for boreholes 6 and 8 is more effective than the line graphs for 2 and 3. Regardless it should be consistent between the boreholes and preferable on one figure to allow for comparison.

I am a little baffled by the discussion of microbial physiology as it relates to sulfur and nitrogen. I am not clear on how high sulfate measurements demands and active sulfur cycle, nor how low dissolved nitrate concentrations preclude an active N cycle. The geochemical data needs to be compared to groundwaters of similar host rock to evaluate the role of water rock interaction in additional to possible microbial processes. If samples still exist, DOC analyses would go a long way to tracking surface input.

Line Comments:

18-20: The two clauses are not clearly linked and it is not clear with more focus on the near surface must be contrasted to study

of unique physiology. Please be more precise.

20-21: Key questions exist (it is not clear what is raising these questions).

26-27: Consider simplifying this sentence.

32-33: Do we know that these microbes originated in the soil, or just that they are similar to common soil stains?

49: relate

55-57: I am not sure that this opening statement is true. We know what a lot about subsurface biogeochemistry and to my knowledge this science has not been clouded by any sort of mania! 57: spanning?

62: I see where you are going, but lost city, is not subsurface as it towers over the ocean floor. Perhaps a different example.

72: Be aware of your subject verb agreement throughout.

76: Further

80-82: I am not sure what this means, please rephrase for clarity

99: word choice. The groundwater is old, but follows reasonably direct flow paths.

96-107: This literature review of past subsurface work needs to be expanded and refined to focus on the knowledge gap that your work is going fill. Studies from some of these places focus on depth gradients and differential impacts of water age and surface inputs.

109-110: I am not sure I am understanding this criticism (or how that will be different in your study). It is not necessary to criticize past work into order to frame your own.

111: most of the waters circulating in the deep subsurface are meteoric in origin (with the exception of some connate brines). I believe you are referring to young meteoric fluids, but need to specify this more directly.

115-116: I am not sure what this means, please clarify

125-126: subject verb agreement

127: As we move into the results I am lacking a description of the history of this mine, where the boreholes came from etc. Is there published work on the hydrology of the system, the age of the rocks? What were they mining? This background is important for understanding your work.

128: periodic, seasonal, occasional, continuous? What sort of recharge are we talking about?

135: a line or two here or in the introduction describing how many boreholes you are working with and how they relate to each other would be helpful.

136: Since the methods come last it would be helpful to say what these are. Major anions and cations, trace elements?

137: what magnitude are the fluctuations compared to analytical precision? 138: Do you mean that each borehole is different?

141: What was the sampling interval?

141: required months or years to refill?

142-143: What are these different than the boreholes in the previous statement?

144: physical parameters and chemical composition

145-146: attributed is interpretation, which should wait for the discussion. However if you mean to say that "Temperatures varied seasonally generally remaining... and reaching.. during the summer" then that is ok.

151: Concentrations of anions?

151-153: I am not sure what this means, please clarify

152-157: For each chemical constituent that is below detection, state the detection limit.

164: are there other values which were measured but not recorded?

166-169: this symbol is the derivative symbol, not the greek letter delta as is appropriate for reporting isotope values.

167: plot above. It would be helpful to plot mean annual and month values of surface precipitation from this area to compare to your subsurface data. Also, it is not clear why resolution of the X and y axes are so different, that makes it difficult to see the slope in the GMWL

169-172: This is interpretation and also, it is not clear how you have ruled out other groundwaters. Please relocate, expand, and clarify.

176: as amplicon sequencing data is non-quantitative, technically your amplicons are dominated by bacteria, not the fluid directly.

179: this is in contrast to your reported dissolved oxygen measurements.

186-187: How does this follow? High sulfate suggests it is present, and enough of it isn't being used for you to measure it. Are you suggesting that the sulfate is produced via sulfide oxidation?

Also, this is interpretation and belongs in a discussion.

186: formatting and charge - that is not sulfate.

194: always say relative abundance when you mean relative abundance.

199: are there actually more other ASVs identified after the firmicutes go down, or can you just see them on the graph better because everything is in relative abundance.

181-208: This paragraph features lots of discussion, quantitative interpretation of relative data, and over interpretation of how far you can use 16S data to imply function. If you have an ASV which is extremely closely related to a strain that does something specific, then state how closely related it is and then interpret (although do so in the discussion).

209-228: Same comment as for the above paragraph. Also, reading between the lines it sounds like these boreholes had lower fluid flow (longer recharge time), but you are interpreting them as being more connected to meteoric input and also are ignoring groundwater sources to recharge? That seems inconsistent to me.

266: rainwater entering the system and reacting with the host rock sufficiently to achieve these chemical compositions, seems extremely rapid. As noted above, this is also not consistent with the microbiology. Please consider a groundwater source.

289-290: I would suggest that DOC analysis would be a very useful means of tracking surficial inputs.

292-293: please avoid hyperbole

294: depleted suggests it was high and now it is low, do you know that?

300-301: This argues against a very young meteoric origin and I am not sure that the cross plot is sufficiently useful to be a main figure in the paper.

368-369: I am not following this conclusion.

462: when were these particular holes drilled..

Staff Comments:

Preparing Revision Guidelines

For complete guidelines on revision requirements, please see the Instructions to Authors at [link to page]. **Submissions of a paper that does not conform to Microbiology Spectrum guidelines will delay acceptance of your manuscript.**

Please return the manuscript within 60 days; if you cannot complete the modification within this time period, please contact me. If you do not wish to modify the manuscript and prefer to submit it to another journal, please notify me of your decision immediately so that the manuscript may be formally withdrawn from consideration by Microbiology Spectrum.

If you would like to submit an image for consideration as the Featured Image for an issue, please contact Spectrum staff.

Review of Thieringer, Honeyman, and Spear “Spatial and temporal constraints on the composition of microbial communities in subsurface boreholes of the Edgar Experimental Mine”

Summary: This manuscript reports on a long-term study of the geochemistry and microbiology of subsurface fluids from boreholes within the Edgar Experimental Mine. The authors observe differences in geochemistry between sites, and concurrent changes in microbial community composition, even at close spatial scales. While the work itself is interesting, the logical framing of key aspects of the work, the scholarship in interpretation of both shallow subsurface geochemistry and microbial physiology, and the writing itself all present major problems which should preclude publication at this time.

Major Comments:

I commend the authors on completing this long-term field study of subsurface microbiology, a rare feat. The study itself is interesting and I think with sufficient rewriting should be publishable. That said, the way this is written and presented is currently ineffective and will require major revisions.

I refrained from line editing this document, but it needs a thorough polish and cut from a senior editor. I am extremely disappointed in the senior author for allowing this to be submitted in its current state. Most of the writing is verbose and includes many clauses where meaning is not clear and subject verb disagreement precludes precise interpretation. There are many superlatives and colloquialisms which interfere with clearly conveying the scientific messages. There are also structural problems, namely, a lack of necessary information in the introduction, lots of interpretation is present in the results, and the discussion is circuitous and repetitive (often repeating parts of the results). When rewriting please try to be more precise with the presentation of the study data and more thorough in referencing information needed for interpretation.

Scientifically there are also some major problems.

The interpretation of fluid origin in contrast to microbial origin is problematic. Most groundwater (even very old ground water) is of meteoric origin, but that doesn't mean it comes in directly from the surface. A groundwater origin is discounted in favor of a meteoric origin (which again is not mutually exclusive), but this interpretation of extremely young fluids is not well supported. It is also not consistent with the surface connectivity or recharge times, or fluid chemistry reported for each borehole and reflected in the microbial diversity data.

Interpretation of physiology from tag sequencing data is always a risky business and must be done with considerable care. To interpret physiology, you must have a very close match with cultured organisms, not just at the family level, but by evaluating your individual ASVs, stating how close they are to organisms with physiology information, and then proceeding with cautious interpretation. Also, looking at relative abundance data means that when one thing goes down, others go up, but that does not mean that the actual abundance of those organisms went up. There is no mention of diversity metrics or other normally ecology statistics which might help get around some of these challenges. I think the display of taxonomic data could be

far better as well. I think the heat map shown for boreholes 6 and 8 is more effective than the line graphs for 2 and 3. Regardless it should be consistent between the boreholes and preferable on one figure to allow for comparison.

I am a little baffled by the discussion of microbial physiology as it relates to sulfur and nitrogen. I am not clear on how high sulfate measurements demands and active sulfur cycle, nor how low dissolved nitrate concentrations preclude an active N cycle. The geochemical data needs to be compared to groundwaters of similar host rock to evaluate the role of water rock interaction in addition to possible microbial processes. If samples still exist, DOC analyses would go a long way to tracking surface input.

Line Comments:

18-20: The two clauses are not clearly linked and it is not clear with more focus on the near surface must be contrasted to study of unique physiology. Please be more precise.

20-21: Key questions exist (it is not clear what is raising these questions).

26-27: Consider simplifying this sentence.

32-33: Do we know that these microbes originated in the soil, or just that they are similar to common soil stains?

49: relate

55-57: I am not sure that this opening statement is true. We know what a lot about subsurface biogeochemistry and to my knowledge this science has not been clouded by any sort of mania!

57: spanning?

62: I see where you are going, but lost city, is not subsurface as it towers over the ocean floor. Perhaps a different example.

72: Be aware of your subject verb agreement throughout.

76: Further

80-82: I am not sure what this means, please rephrase for clarity

99: word choice. The groundwater is old, but follows reasonably direct flow paths.

96-107: This literature review of past subsurface work needs to be expanded and refined to focus on the knowledge gap that your work is going fill. Studies from some of these places focus on depth gradients and differential impacts of water age and surface inputs.

109-110: I am not sure I am understanding this criticism (or how that will be different in your study). It is not necessary to criticize past work into order to frame your own.

111: most of the waters circulating in the deep subsurface are meteoric in origin (with the exception of some connate brines). I believe you are referring to young meteoric fluids, but need to specify this more directly.

115-116: I am not sure what this means, please clarify

125-126: subject verb agreement

127: As we move into the results I am lacking a description of the history of this mine, where the boreholes came from etc. Is there published work on the hydrology of the system, the age of the rocks? What were they mining? This background is important for understanding your work.

128: periodic, seasonal, occasional, continuous? What sort of recharge are we talking about?

135: a line or two here or in the introduction describing how many boreholes you are working with and how they relate to each other would be helpful.

136: Since the methods come last it would be helpful to say what these are. Major anions and cations, trace elements?

137: what magnitude are the fluctuations compared to analytical precision?

138: Do you mean that each borehole is different?

141: What was the sampling interval?

141: required months or years to refill?

142-143: What are these different than the boreholes in the previous statement?

144: physical parameters and chemical composition

145-146: attributed is interpretation, which should wait for the discussion. However if you mean to say that "Temperatures varied seasonally generally remaining... and reaching.. during the summer" then that is ok.

151: Concentrations of anions?

151-153: I am not sure what this means, please clarify

152-157: For each chemical constituent that is below detection, state the detection limit.

164: are there other values which were measured but not recorded?

166-169: this symbol is the derivative symbol, not the greek letter delta as is appropriate for reporting isotope values.

167: plot above.

It would be helpful to plot mean annual and month values of surface precipitation from this area to compare to your subsurface data. Also, it is not clear why resolution of the X and y axes are so different, that makes it difficult to see the slope in the GMWL

169-172: This is interpretation and also, it is not clear how you have ruled out other groundwaters. Please relocate, expand, and clarify.

176: as amplicon sequencing data is non-quantitative, technically your amplicons are dominated by bacteria, not the fluid directly.

179: this is in contrast to your reported dissolved oxygen measurements.

186-187: How does this follow? High sulfate suggests it is present, and enough of it isn't being used for you to measure it. Are you suggesting that the sulfate is produced via sulfide oxidation?

Also, this is interpretation and belongs in a discussion.

186: formatting and charge – that is not sulfate.

194: always say relative abundance when you mean relative abundance.

199: are there actually more other ASVs identified after the firmicutes go down, or can you just see them on the graph better because everything is in relative abundance.

181-208: This paragraph features lots of discussion, quantitative interpretation of relative data, and over interpretation of how far you can use 16S data to imply function. If you have an ASV which is extremely closely related to a strain that does something specific, then state how closely related it is and then interpret (although do so in the discussion).

209-228: Same comment as for the above paragraph. Also, reading between the lines it sounds like these boreholes had lower fluid flow (longer recharge time), but you are interpreting them

as being more connected to meteoric input and also are ignoring groundwater sources to recharge? That seems inconsistent to me.

266: rainwater entering the system and reacting with the host rock sufficiently to achieve these chemical compositions, seems extremely rapid. As noted about, this is also not consistent with the microbiology. Please consider a groundwater source.

289-290: I would suggest that DOC analysis would be a very useful means of tracking surficial inputs.

292-293: please avoid hyperbole

294: depleted suggests it was high and now it is low, do you know that?

300-301: This argues against a very young meteoric origin and I am not sure that the cross plot is sufficiently useful to be a main figure in the paper.

368-369: I am not following this conclusion.

462: when were these particular holes drilled.

Reviewer #1:

Major Comments:

The manuscript "Spatial and temporal constraints on the composition of microbial communities in subsurface boreholes of the Edgar Experimental Mine" by Theieringer et al. seeks to quantify the microbial community composition within fluids emanating from experimental boreholes at the mine. Using mines to access subsurface microbes is an excellent and cost-effective way to investigate the deep biosphere. The authors use an ingenious, simple packing system to isolate the boreholes and to collect fluids. Overall, I thought the manuscript needs condensing, especially the introduction. Additionally, I'm concerned that the authors have not properly analyzed their dataset correctly. The methods are sparse and so it is difficult to assess what they have actually done with the amplicon reads. DADA2 has lots of nuances and caveats that can lead to screening of sequences and so I'm worried that one reason the number of reads/sample is so low, is that DADA2 is screening out real data. I've highlighted this down below. Furthermore, it seems that kit contamination may be a problem, due to low biomass of their samples. There are several papers describing this and one recent one for subsurface systems (see below). Please be aware that this can be a large problem for subsurface systems. I think this is an interesting system that should be highlighted but your narrative needs focus to really bring that out

We have addressed the major concerns of the reviewer in full in their line-by-line comments. We have made serious revisions to the introduction to condense the verbosity of the manuscript. Additionally, we have removed unnecessary points of discussion in the results section that were repetitive. We have provided a major refocus of the results section to highlight significant data that was not well established in the original line graph figures. We have updated the microbial community analysis to include additional heatmap figures for simplicity of observing the changes in microbial composition over the course of the experiment.

We have taken deep consideration to address all the concerns regarding the analysis of our sequencing data and have included the necessary information in our methods section where it was lacking. We addressed the concern of running our sequencing data at two different facilities in the line-by-line comments. In brief, we ran DADA2 independently on each of the bulk sequencing runs and then used the "mergeSequenceTables" function before removing chimeras and further downstream analysis. There were no observable changes in our sequencing analysis from this method. We have also taken into consideration the potential problem for kit contamination raised by the reviewer. We followed the steps outlined in Sheik et al. (2018) to use a highly sensitive approach to remove potential contamination by including lab blanks in our sequencing analysis. We removed specific ASVs determined from lab blanks to remove the likelihood of kit contamination effecting our sequencing analysis.

Introduction

Introduction thoughts: I think the intro needs some tightening and reworking. All the ideas are there but could be presented more succinctly.

We have taken this comment into deep consideration and removed verbose sentences in order to provide a more tightknit introduction. Our only addition to the introduction includes information regarding the brief history of the mine and more detail about the boreholes being observed in this study.

[L 117-120: Once an active economic mining site, it was repurposed for teaching demonstrations by Colorado School of Mines for mining engineering practices. Four boreholes drilled from previous mining teaching practices are the focus of this study, located centimeters to meters apart along the same wall of our field site.]

L 59: "analogs to.." instead of analogous to...

Thank you for pointing this typo out. It has been addressed and fixed.

L60-64: The focus on marine systems here seems out of place. I understand what you are going for but still seems out of place or maybe needs to be reworded a bit.

The focus of this section was to highlight the amount of research that has come from marine systems in elucidating the limits of microbial life. The intent was to provide additional support and comparison for the advancement of research focused on terrestrial systems. Following both reviewers suggestions, we have removed this section of the manuscript to cut down the verbosity of the introduction. Additionally, it detracts from jumping straight into discussing the similarity of relevant terrestrial systems.

L 69: Fount seems to be the wrong word here. I take it to mean source but seems strange. Maybe biodiversity would work better?

Thank you for addressing the confusion of this word. We have changed the line to include the word "biodiversity" in order to read more clearly and directly relate to the structure of the sentence.

L 80: I don't think the sources energy of chemolithoautotrophy are unique in hydrothermal settings. You still see the same redox couplings in terrestrial systems both in the deep subsurface, vadose zones, and surface. Maybe reword?

We agree with the reviewer's assessment that the same redox couplings and chemolithotrophic metabolisms are found in variety of environments, regardless if they are extreme or nutrient dense systems. The sentence has been rephrased to address that the sources of energy sustaining microbial communities in subsurface systems are not necessarily different from the zones highlighted by the reviewer, but are instead more difficult to attain in nutrient-limiting and low biomass systems. The line has been restructured as such:

[L 72-75: Even more, water-rock interactions demonstrated in terrestrial subsurface environments distinguish the chemical disparity of chemolithotrophic processes that sustain ecosystems of low biomass utilizing restricted sources of energy [15,16].]

L91-95: This seems out of place and somewhat off topic as you introduce the paragraph discussing nutrients and then talk solely about water and not the nutrients that are dissolved.

We agree with the reviewer in that this section of the introduction is verbose and has been removed in order to provide more tightening.

L96: another laboratory to look at is Soudan Iron Mine, see Sheik 2021 Frontiers.

Thank you for providing this additional resource. We have reviewed the paper and included it in our review of subsurface observatories.

[L 94-97: Microorganisms that exist in the secluded hematite iron formation of the Soudan Underground Mine State Park were examined with metagenomic analyses to determine genes representing diverse metabolic strategies from a low biomass system to overcome nutrient limitations [21].]

Results

L154: when discussing ions avoid using levels. You are talking about the chemistry of the system and so use concentration instead.

We have resolved the issue of incorrect phrasing related to describing chemical values and have replaced all phrasing throughout the manuscript to reflect the correct terminology.

L 149-165: While everyone should be accustomed to element's shorthand you should probably still use the full word, like Nitrate (NO₃⁻) or nitrite (NO₂⁻). Also this section could also be condensed a bit.

We have replaced all shorthand notation with the full word for different analytes in the manuscript. Additionally, this section has been reworded in order to address the geochemistry table which we have included in the supplementary information. We included information regarding the PCA analysis in order to better lead to the conclusion of delineating between metal-rich and depleted fluids that can be diagnostic of the microbial community differences seen in the borehole fluid samples.

Figure 2/table2: You have a lot of Geochem data here in Table 2. Maybe put this all in supplemental and use a RDA style analysis to show how the boreholes relate to each other and which elements/anions are significantly delineating the boreholes. Its unclear why you would focus only on Zn vs Mn

We accept this suggestion and have moved the geochemistry table into the supplemental information along with the detection limits. We have included a PCA style analysis to show how the geochemistry between the boreholes relate and replaced the original Figure 2.

We focus on Zn and Mn specifically because of the relation to potential ore bodies found from economic mining. They can suggest where metal rich fluid can exist that has equilibrated with the surrounding host rock and in turn influenced the composition of the microbial community that may exist. Similarly, this figure demonstrates a conserved relationship of these two metals from individual boreholes despite sampling period or frequency.

L 166: This may be just my pet peeve but starting out a paragraph with "Figure 3" seems odd. I think rephrasing this sentence to frame what and why you are measuring would be better.

We have rephrased this sentence to clarify why we measured water isotopes and their relation to other subsurface systems.

[L 166-170: The $\delta^{18}\text{O}$ and $\delta^2\text{H}$ signatures for borehole fluids are compared to the Global Meteoric Water Line (GMWL) along with samples from other subsurface systems or observatories (**Figure 3**). This trendline serves as an investigative tool for understanding if the subsurface fluids have undergone some extent of low-temperature water-rock interactions or remain isotopically conserved to reflect meteoric recharge from the surface [17,19].]

Figure 3: I think you need to plot some of the other subsurface values for some of the other mines you mention in the introduction. Place your values into context with other mines/study sites.

We have updated Figure 3 to include values from some of the other subsurface studies we have mentioned in the introduction. This should provide a more clear context of the borehole fluids from our study against other (deeper) subsurface systems where fluids are much older and have undergone a greater degree of residence time in the subsurface to deviate more strongly from the GMWL.

Microbial analysis

L 181-185: It is difficult/impossible to see any of this data since borehole 2 is only shown at the phylum level. You also have a lot of discussion happening in your results section. Either combine into a results/discussion or move to discussion

This figure and section have been updated accordingly. All representation of taxonomy is represented in a heatmap for all boreholes and soil samples. This section of the results has also been amended to remove any discussion and been reformatted.

Figures 4/5/6: While this is a time series, using line graphs to depict this type of data is not great. I think showing us a stacked bar graph of all the samples would perhaps be more useful to see how changes are occurring relative to one another. Perhaps just expand figure 6 to include all the boreholes.

We have addressed this concern by replacing figures 4 and 5 with a heatmap of the relative abundance, and included the soil samples in the original heatmap of Boreholes 6 and 8. The order of figures has been restructured and they now are figures 6 and 7. Additionally, the heatmaps include the top 15 most abundant genera in order to provide more detail of the composition of the microbial community that exists in each borehole over the course of the sampling experiment. This additionally provides a side-by-side contrast of the community composition in Boreholes 6 and 8 to the soils to allow us to better assess the relationship of potential surface input into the subsurface. We have addressed our reevaluation in the results and discussion section.

Observation: I think starting with the PCoA results would better set up your taxonomic analysis of the boreholes. Establishing there is a difference and then telling us what is driving the difference would be good.

We have taken this comment into consideration and restructured the results and discussion section to begin with the PCoA analysis to set up our microbial community analysis of the boreholes. We then take a deeper dive into the specific taxa present afterwards to explain potential drivers of the differences we observe.

Taxonomy: You are using a lot of handwaving in this section and combining discussion. If that is your intent change to a combined results/discussion. Did you do any statistical analysis like LefSE or random forest modelling to pull out taxa that are significantly different in the boreholes? Feels like that would be a better approach here.

We have conducted a SIMPER analysis in order to assess the taxa that explain the greatest contribution of difference between the boreholes. We have removed any interpretation from the

results section and included our SIMPER analysis in the discussion to reflect significantly important taxa that are representative within each borehole.

You mention that these are all soil-related groups. Is there any direct sharing of sequences that are from the soil and found in the subsurface samples?

We have readdressed our analysis of the microbial community composition in Boreholes 6 and 8 after greater scrutiny of the sequences observed in our soil samples. *Bradyrhizobium* is the dominant taxa shared between the soil and subsurface borehole samples. However, we have taken further steps mentioned in the comments to address potential kit contaminants and believe that the presence of additional taxa within Boreholes 6 and 8 still do not represent an endemic subsurface microbial community, but rather one that has been influenced by some surficial transport or human interaction during the initial installment of the packers. We have discussed this in detail in the manuscript in our revised results and discussion sections.

Discussion:

In your methods you don't mention that you screened your sequences from kit contaminants. Sheik et al 2018 show that for low biomass subsurface systems this is necessary and also list common genera that need to be screened and/or highly scrutinized. Your abundance of "soil genera" in your low-flow boreholes would suggest that kit contamination may be a large problem. I would definitely suggest being very careful with your taxonomic analysis.

We appreciate these comments and have addressed this issue in full. We have made sure to include information in our methods section regarding how we screened for kit contamination that may have influenced the outcomes of our sequencing analysis. We have followed the instructional flowchart that is described in the paper mentioned (Sheik et al 2018), by including "lab blanks" used during our DNA extraction procedure to follow the "Highly Sensitive" removal process in order to identify potential contaminants within our samples.

You say that there is anaerobic chemolithoautotrophy several times. However, your O₂ in the waters is quite high in every borehole. If you have any reduced fluids that are interacting abiotic oxidation is going to be a problem, especially with sulfur and definitely with iron.

We have addressed comments regarding anaerobic chemolithotrophy that are egregious and do not match the findings we have displayed. We have reformatted the discussion to reflect the data appropriately and such comments have been rewritten.

Methods:

In general, your methods are very sparse. For instance, to generate a Unifrac metric, you need a phylogenetic tree. How did you produce this? You say you used a modified pipeline and then cite a paper. Did you use their pipeline they developed for ASVs generation or did you modify theirs in some way? These are all very important details that need to be in the paper.

We have amended this methods section in order to include how we generated ASV's and a phylogenetic tree. We also clarified any misconceptions of the pipeline we used in order to

evaluate our sequences. We followed the DADA2 pipeline in order to create a phyloseq object, and we used the “phangorn” package in order to create a phylogenetic tree.

[L 544-550: The resulting FASTQ files were demultiplexed and trimmed with AdapterRemoval2 [61]. DADA2 was used to filter reads by error rates, amplicon sequence variants (ASVs) were identified, paired-end reads were merged to construct a sequence table, and chimeric sequences were removed[62]. Taxonomic assignments were called with the SILVA SSU database training file (version 138). A phylogenetic tree was created with the “phangorn” package in R by first constructing a neighbor-joining tree then fitting a generalized time-reversible with Gamma rate variation maximum likelihood tree [63].]

You're using two different places to sequence, did you send this off samples multiple times at each facility or was it all just 2 bulk runs? In either case you need to specify if you ran DADA2 independently on each run then combined all the runs together with DADA at the end to create an ASV table. This will definitely have downstream effects with the model prediction for the ASVs, since it is well documented that there are inconsistencies between sequencing facilities.

We have addressed this miscommunication of producing an ASV table. We did run DADA2 independently on each of the bulk sequencing runs and then used the “mergeSequenceTables” function before beginning the step of removing chimeras and further downstream analysis of generating a taxonomy table, phylogenetic tree, and ASV table. We did not detect any observable changes in our sequencing analysis.

Why did you get such low recovery of ASVs? With MiSeq you should be getting well over 15K Seqs per sample. Are these being screened out during the ASV creation? The chimera removal step from Dada2 is very sensitive and can screen out real sequences. Did you check to see if this is happening? See above for some reasons why. But also how did you remove your primers/adapters?

We have mentioned in our methods section that we used “AdapterRemoval2” in order to remove our primers. We have checked our chimera removal step and retained 94% of our sequences. The waters sampled in this study are particularly low biomass, and due to the constraints of the borehole volume and recharge capabilities we were only able to filter as little as ~100 mL of water; this can be much greater in other subsurface systems. Although we manually normalize PCR products by concentration prior to MiSeq library preparation, it is likely not possible to fully compensate for having little sample DNA to begin with. One possibility would be to increase the number of PCR cycles for low-biomass samples, but we choose not to do this due to the spurious nature of some amplicons when the number of PCR cycles becomes too numerous.

L 542: I think the name of the Duke Center you used is incomplete or is mistyped

We have fixed this error to contain the complete name of the sequencing facility.

Figures: Be cognizant of what your figures are trying to convey. You have lots of data but the figures don't necessarily show this. Please use different shapes for each borehole and use colors that are color-blind appropriate. Several of the colors bleed into each other making it tough to discern which borehole/location they belong to.

We have addressed this comment by retaining the shapes and colors of sample data throughout figures presented in this revised manuscript in order to create consistency of the boreholes represented in all of the data. The previous color palette used was one version of colorblind-

friendly, but we have now selected a different one that does not bleed between data points as the reviewer observed.

Reviewer #2:

Major Comments:

I commend the authors on completing this long-term field study of subsurface microbiology, a rare feat. The study itself is interesting and I think with sufficient rewriting should be publishable. That said, the way this is written and presented is currently ineffective and will require major revisions. I refrained from line editing this document, but it needs a thorough polish and cut from a senior editor. Most of the writing is verbose and includes many clauses where meaning is not clear and subject verb disagreement precludes precise interpretation. There are many superlatives and colloquialisms which interfere with clearly conveying the scientific messages. There are also structural problems, namely, a lack of necessary information in the introduction, lots of interpretation is present in the results, and the discussion is circuitous and repetitive (often repeating parts of the results). When rewriting please try to be more precise with the presentation of the study data and more thorough in referencing information needed for interpretation.

We have taken these comments into deep consideration and heavily revised the manuscript to avoid superfluous language. We have detailed comments addressing the line-by-line suggestions made by the reviewer to remove passages that were confusing or created structural problems. We condensed the introduction to remove excessive description to set up our study. However, we provided additional background information (primarily about the Edgar Experimental Mine) to allow the reader to have a greater understanding of the system at the beginning of the manuscript. We have addressed the circuitry of the results and discussion sections with major rewriting of the results – especially after deeper analysis from both reviewers and reframing how the data is presented. The results are reformatted to remove any interpretation, and the discussion is more succinct in elaborating on the results of the study. Any specific comments are addressed in full within the line-by-line comments.

Scientifically there are also some major problems.

The interpretation of fluid origin in contrast to microbial origin is problematic. Most groundwater (even very old ground water) is of meteoric origin, but that doesn't mean it comes in directly from the surface. A groundwater origin is discounted in favor of a meteoric origin (which again is not mutually exclusive), but this interpretation of extremely young fluids is not well supported. It is also not consistent with the surface connectivity or recharge times, or fluid chemistry reported for each borehole and reflected in the microbial diversity data.

We have acknowledged this criticism and reframed our outlook of the origin of the fluid source feeding the boreholes at Edgar Mine. We included additional isotopic data from other deep subsurface sites to provide a reference for our $\delta^{18}\text{O}$ and $\delta^2\text{H}$ results. This provides a framework to demonstrate that the fluid trapped within the boreholes of the Edgar Mine have not undergone any extensive water-rock interactions requiring long spans of geologic time to alter the isotopic signature of the water. Instead, the fluid within the boreholes is relatively young in geologic age, and is interpreted to be of meteoric origin due to our isotopic data falling so closely on the global meteoric water line. Additionally, the physical location of the Edgar Mine is above the town of Idaho Springs and Clear Creek Canyon, and would suggest that a deeper and older groundwater source feeding the boreholes is unlikely.

Interpretation of physiology from tag sequencing data is always a risky business and must be done with considerable care. To interpret physiology, you must have a very close match with cultured organisms, not just at the family level, but by evaluating your individual ASVs, stating how close they are to organisms with physiology information, and then proceeding with cautious interpretation. Also, looking at

relative abundance data means that when one thing goes down, others go up, but that does not mean that the actual abundance of those organisms went up. There is no mention of diversity metrics or other normally ecology statistics which might help get around some of these challenges. I think the display of taxonomic data could be far better as well. I think the heat map shown for boreholes 6 and 8 is more effective than the line graphs for 2 and 3. Regardless it should be consistent between the boreholes and preferable on one figure to allow for comparison.

We have taken these comments into deep consideration along with the other reviewer's comments about assessing our sequencing data. We have provided a clearer and more effective method for presenting our taxonomic data by replacing the original line graphs with additional heatmaps for all of the boreholes. Additionally, we have included the taxonomic data of the soil samples to provide direct comparison to the boreholes. We provide careful detail in assessing the role of the genus-level organisms we have identified in the borehole fluids to ensure a fair evaluation of potential metabolic capabilities as it is compared to our geochemical data. We have also addressed any miscommunication in defining the relative abundance of our amplicon sequencing data in relation to the observed microbial members within the borehole fluids.

I am a little baffled by the discussion of microbial physiology as it relates to sulfur and nitrogen. I am not clear on how high sulfate measurements demands and active sulfur cycle, nor how low dissolved nitrate concentrations preclude an active N cycle. The geochemical data needs to be compared to groundwaters of similar host rock to evaluate the role of water rock interaction in addition to possible microbial processes. If samples still exist, DOC analyses would go a long way to tracking surface input.

We have addressed this critique in full to reconsider how we evaluate the microbial activity within the near-subsurface of the Edgar Mine. We unfortunately do not have additional samples to measure DOC, but is suggested for future work and analysis to guide further interpretation of the biogeochemical cycling within the mine. We removed any confusing language within the manuscript that would conclude or demand any interpretation of active sulfur or nitrogen cycling. We combine our geochemical data that demonstrates high concentrations of sulfate with prominent sulfur cycling microbiota that are prominent within Boreholes 2 and 3. It is then suggested that there may be a role for these microorganisms to play role within these borehole fluids due to their large relative abundance. Similarly, we observed members such as *Candidatus Nitrotoga* which can adapt to nitrite limited conditions and perform sulfur oxidizing metabolic capabilities. Our geochemical data demonstrates largely undetected amounts of nitrite, which could suggest that limited nitrite is available for microbial metabolisms. While undetected amounts of an analyte do not indicate a lack of available nutrients for microorganisms to use within the subsurface, it does suggest that nitrogen may be a limiting nutrient with less availability.

Introduction

18-20: The two clauses are not clearly linked and it is not clear with more focus on the near surface must be contrasted to study of unique physiology. Please be more precise.

We acknowledge the difficulty in clarity of this sentence and have re-worded the sentence as such:

[L 20-22: Attention is required to understand the near-subsurface and its continuity with surface systems, where numerous novel microbial members with unique physiological modifications remain to be identified.]

20-21: Key questions exist (it is not clear what is raising these questions).

We added to the sentence in order to clarify that the interaction between subsurface and surface processes is what is driving these questions:

[L22-25: This surface-subsurface relationship raises key questions about networking of surface hydrology, geochemistry affecting near-subsurface microbial composition, and resiliency of subsurface ecosystems.]

26-27: Consider simplifying this sentence.

We acknowledge the reviewers comment to simplify the sentence. However, we believe that the sentence contains relevant information regarding what was sampled, the complete time of the experiment, where samples were collected from, the spatial resolution of the sampling devices, and what was being observed of samples all in a single sentence. We have chosen to leave the sentence unedited to contain vital information of the study in a succinct sentence for the abstract.

32-33: Do we know that these microbes originated in the soil, or just that they are similar to common soil stains?

We have rephrased the sentence for clarity as to not assume a direct origination from the soils, but highlight the overlap of common soil microbial taxa that are shared in some of the borehole fluids.

49: relate

We have fixed the subject-verb agreement of this word to correctly fit the sentence structure.

55-57: I am not sure that this opening statement is true. We know what a lot about subsurface biogeochemistry and to my knowledge this science has not been clouded by any sort of mania!

We acknowledge the point the reviewer makes here about the word choice in this sentence. It appears that “fervor” is being misconstrued for the message this sentence is trying to convey. We have rephrased the sentence for clarity. We are not denying that there is plenty of knowledge constraining subsurface processes and the biogeochemistry of many subsurface systems. Instead, we are highlighting that the true diversity, abundance, and knowledge of microorganisms in the subsurface still requires much work, as it is what motivated our investigation.

[L 57-61 : The intricacies of the deep subsurface have triggered many investigations into the microbial habitability sustaining itself in the rock-hosted subsurface. The chemical drivers that govern the transition from terrestrial surface to subsurface life are not fully understood – especially in near-subsurface zones where the system may behave as a hybrid between surface and subsurface processes.]

57: spanning?

We have replaced the word choice suggested by the reviewer in the sentence to provide more clarity.

62: I see where you are going, but lost city, is not subsurface as it towers over the ocean floor. Perhaps a different example.

We have removed this section from the introduction in order to be more succinct and straightforward about terrestrial subsurface systems as they relate to our experiment. We do wish to acknowledge that there has been research conducted at the Lost City Hydrothermal Field at the vents, but also within the subsurface from cores taken at the Atlantis Massif which hosts the field site as highlighted by Lang and Brazelton 2020 and Früh-Green et al. 2018.

72: Be aware of your subject verb agreement throughout.

This has been addressed in the manuscript throughout and revised for further grammatical error.

76: Further

We have accepted this correction and edited the sentence for word choice.

80-82: I am not sure what this means, please rephrase for clarity

We intend to highlight that the common perception of an “extreme” environment is usually associated with an extreme environmental condition – this could be temperature, pH, pressure. We then suggest that limited nutrient availability does not get characterized in this “extreme” definition, but should be considered as such.

99: word choice. The groundwater is old, but follows reasonably direct flow paths.

We accept this error in word choice and have removed the word entirely for the information to read more clearly.

96-107: This literature review of past subsurface work needs to be expanded and refined to focus on the knowledge gap that your work is going fill. Studies from some of these places focus on depth gradients and differential impacts of water age and surface inputs.

We acknowledge the reviewer’s critique, however, we believe that our review of the subsurface observatories displays how many subsurface studies take place within deep subsurface systems kilometers below the surface or examining extremely old fluid sources. This sets up our study which takes place in the shallow subsurface and looks at direct surface interaction from young meteoric water that infiltrates into our mine system. The knowledge gap that we are addressing is the lack of studied systems that are near-subsurface in nature as likely these systems will display unique behavior relative to their deep and / or terrestrial counterparts

109-110: I am not sure I am understanding this criticism (or how that will be different in your study). It is not necessary to criticize past work into order to frame your own.

We have reworked the sentence to be more concise and clearer. We intend to highlight what is previously mentioned from our review of the subsurface observatories: that not as much work has been conducted regarding the interaction of young meteoric fluids with shallow subsurface environments. We have edited the sentence to read as follows:

[L 99-101: However, the scope of which microorganisms survive in shallow subsurface environments from young meteoric water infiltration to introduce nutrients from the surface versus chemical energy sources originating in local settings remains insufficiently answered [17].]

111: most of the waters circulating in the deep subsurface are meteoric in origin (with the exception of some connate brines). I believe you are referring to young meteoric fluids, but need to specify this more directly.

We have addressed this comment by referring to the meteoric water as “young.” We have additionally addressed this throughout the manuscript where it is not directly stated.

115-116: I am not sure what this means, please clarify

We agree with this comment that the sentence does not read clearly. We have eliminated the sentence to provide a more succinct introduction. Additionally, it is addressed in the final paragraph of the introduction where it more appropriately fits into the description of the experiment with an additional sentence to clarify how multiple opportunities to sample allow for a temporal investigation of the microbial community within the mine.

[L 121-124: Meteoric groundwater infiltrating the fractures of the mine leak out of the boreholes, offering the opportunity to study the subsurface microbiology of meteoric water influenced by young fluid-rock equilibration (**Figure 1**). Additionally, the periodic (~2 weeks) to seasonal recharge of water makes it possible to study temporal influences through incubation and recharge periods.]

125-126: subject verb agreement

We have addressed and fixed the subject verb agreement of “opportunity” to “opportunities.”

127: As we move into the results I am lacking a description of the history of this mine, where the boreholes came from etc. Is there published work on the hydrology of the system, the age of the rocks? What were they mining? This background is important for understanding your work.

We provide a more detailed description of the mine and its history in our methods section within the Field Site subsection. The boreholes are a result of educational demonstrations/programs within the mine where students practice drilling techniques related to mining engineering. The rocks are of Precambrian age, however, no detailed hydrological studies have been published related to Edgar Experimental Mine. The mine originally produced silver, gold, lead, and copper starting in the 1870's. The context of the mine as a field site is put into greater perspective in the methods section – additionally it cuts down verbosity in the introduction. However, to provide a brief background for context, we added an additional sentence to the final paragraph.

[L 117-119: Once an active economic mining site, it was repurposed for teaching demonstrations by Colorado School of Mines for mining engineering practices.]

128: periodic, seasonal, occasional, continuous? What sort of recharge are we talking about?

We have addressed this comment to include the correct time frame of recharge periods for the boreholes from the mine. This is discussed in the very beginning of the results section of the manuscript.

[L 123-124: Additionally, the periodic (~2 weeks) to seasonal recharge of water makes it possible to study temporal influences through incubation and recharge periods.]

135: a line or two here or in the introduction describing how many boreholes you are working with and how they relate to each other would be helpful.

We acknowledge this lack of information and have addressed the comment by adding a brief sentence to give context to the number of boreholes in our study and how they relate spatially. [L 119-120: Four boreholes drilled from previous mining teaching practices are the focus of this study, located centimeters to meters apart along the same wall of our field site.]

Results

136: Since the methods come last it would be helpful to say what these are. Major anions and cations, trace elements?

We have fixed the issue and have rephrased the sentence to read as follows: [L 131-133: While boreholes minorly fluctuate in analyte (major anion and cation) concentration, no discrete patterns were observed.]

137: what magnitude are the fluctuations compared to analytical precision?

All of the fluctuations are within the same order magnitude of change compared to the analytical precision for both IC (0.1 mg/L for all and 0.5 mg/L for Phosphate) and ICP (see supplementary information). The only outlier would be chloride, which changes over two orders of magnitude in concentration, but does not show any consistency in fluctuation with other boreholes or based upon date of extraction. We have moved the table of geochemistry to the supplementary information and included the detection limit for each anion and cation.

138: Do you mean that each borehole is different?

The sentence is highlighting that the borehole fluid geochemistry may not differ greatly between boreholes or during the overall course of the experiment. However, the geochemistry within each borehole is slightly variable that may be enough to dictate micro-environments for microbial community composition.

141: What was the sampling interval?

The sampling intervals were both predetermined and spontaneous due to the unpredictability of some boreholes recharging with enough fluid for sampling. This is discussed in greater detail within the methods section. In brief, a period of 2-week sampling intervals were chosen to investigate how scheduled incubation periods might affect the microbial community observed. This was primarily targeted at Borehole 2 – the only borehole to fill consistently with enough fluid for sampling extraction. Other boreholes were unpredictable with their fluid recharge, therefore despite arranging dates to collect samples from all of the boreholes, the remaining boreholes filled at inconsistent rates. This results in an uneven sampling interval, but cannot be established prior to visiting the field site.

141: required months or years to refill?

We accept this word choice amendment and have corrected the sentence.

142-143: What are these different than the boreholes in the previous statement?

We have clarified the sentence to reflect the boreholes from the previous sentence. This sentence is referring again to all of the boreholes besides Borehole 2.

144: physical parameters and chemical composition

We have changed the word choice as suggested.

145-146: attributed is interpretation, which should wait for the discussion. However if you mean to say that "Temperatures varied seasonally generally remaining... and reaching.. during the summer" then that is ok.

The sentence has been reworded to avoid misconception with the discussion section.

[L 140-141: Water temperature ranges varied seasonally, generally fluctuating between 10.6 to 13.5°C and reaching 17.6°C in summer.]

151: Concentrations of anions?

We have changed the sentence to be more concise and reads as follows:

[L 146-148: Concentrations of anions (putative electron acceptors for microbial growth as well as phosphorous sourcing) tend to be very low or undetected, allowing for selective substrate availability of microbial communities.]

151-153: I am not sure what this means, please clarify

This sentence is stressing the low and/or undetectable concentrations of anions within the borehole fluids. This can result in limited substrate availability to microbial communities existing within the boreholes that will select for specific metabolisms or require flexible metabolic capabilities of the microorganisms that exist in the borehole fluids. We have changed the word choice to say “selective” instead of niche for future clarification.

152-157: For each chemical constituent that is below detection, state the detection limit.

We have moved the table of geochemistry to the supplementary information and included the detection limit information for each anion and cation.

164: are there other values which were measured but not recorded?

All measured values are included in our table of recorded field, IC, and ICP data. This has been organized into supplementary information for easier viewing and reading, as well as to have all the information collected more accessibly onto a single document. There are not values that were measured but not recorded. We have conducted a PCA style analysis to show where the boreholes are trending regarding the geochemistry. We have highlighted manganese and zinc specifically because they denote where metal-rich fluids can be an indicator for distinguishing associated microbial communities.

166-169: this symbol is the derivative symbol, not the Greek letter delta as is appropriate for reporting isotope values.

We have corrected all of the symbols to be the correct Greek letter for delta.

167: plot above. It would be helpful to plot mean annual and month values of surface precipitation from this area to compare to your subsurface data. Also, it is not clear why resolution of the X and y axes are so different, that makes it difficult to see the slope in the GMWL

We have added additional data from other mines and subsurface observatories to contrast the values reported from this study.

169-172: This is interpretation and also, it is not clear how you have ruled out other groundwaters. Please relocate, expand, and clarify.

We have relocated this sentence to the discussion to reflect interpretation. The water source for Edgar Mine is most likely of meteoric origin, as the $\delta^{18}\text{O}$ and $\delta^2\text{H}$ values will display values similar to the global meteoric water line that reflect the conditions of recharge from surface input over long periods of time unless major water-rock interactions are involved. Therefore, this isotopic tool cannot be used to date the water, but act as an initial proxy for understanding the degree of residence time over shorter geological timespans. It is very likely that the borehole fluids reflect a young meteoric groundwater, however, no interpretation of exact dating can be established. Adding to even greater complexity, the mechanism for determining how water flows through veins and pores in the subsurface rock at Edgar Mine makes it difficult to determine how meteoric groundwater may leak out of certain boreholes more than others, or which boreholes may share a more direct link to surficial inputs with shorter residence times.

[L 166-170 : The $\delta^{18}\text{O}$ and $\delta^2\text{H}$ signatures for borehole fluids are compared to the Global Meteoric Water Line (GMWL) along with samples from other subsurface systems or observatories (**Figure 4**). This trendline serves as an investigative tool for understanding if the subsurface fluids have undergone some extent of low-temperature water-rock interactions or remain isotopically conserved to reflect meteoric recharge from the surface [17,19].]

176: as amplicon sequencing data is non-quantitative, technically your amplicons are dominated by bacteria, not the fluid directly.

We have rephrased the sentence for greater clarity.

[L 177-179: The fluid samples were dominated by *Bacteria* (98.5%) with only a small presence of *Archaea* (1.5%) belonging predominantly to samples from Borehole 2.]

179: this is in contrast to your reported dissolved oxygen measurements.

To avoid interpretation saved for the discussion section, we have removed the comment “indicating anaerobic conditions” and addressed the presence of detailed members in the microbial communities of the boreholes later in the discussion.

186-187: How does this follow? High sulfate suggests it is present, and enough of it isn't being used for you to measure it. Are you suggesting that the sulfate is produced via sulfide oxidation? Also, this is interpretation and belongs in a discussion.

We have moved this section into the discussion as it reflects interpretation. We have clarified the passage to reflect the prominence of sulfate in the fluids and how this can serve as an important electron acceptor for microorganisms, as well as the potential that a portion of this analyte is produced biologically.

186: formatting and charge - that is not sulfate.

We have replaced all abbreviations of cations and anions to their full written notation within the manuscript.

194: always say relative abundance when you mean relative abundance.

We have addressed this issue and fixed any misconceptions in the manuscript.

199: are there actually more other ASVs identified after the firmicutes go down, or can you just see them on the graph better because everything is in relative abundance.

We have included a new heatmap of Boreholes 2 and 3 to demonstrate the flux of relative abundance in microorganisms over the sampling period of this experiment. There is certainly a greater degree of taxa represented in the relative abundance as *Firmicutes* relative abundance goes down, while the inverse occurs as the abundance increases.

181-208: This paragraph features lots of discussion, quantitative interpretation of relative data, and over interpretation of how far you can use 16S data to imply function. If you have an ASV which is extremely closely related to a strain that does something specific, then state how closely related it is and then interpret (although do so in the discussion).

This paragraph has been moved to the discussion in order to address potential metabolic functions of the microbial populations within the boreholes.

209-228: Same comment as for the above paragraph. Also, reading between the lines it sounds like these boreholes had lower fluid flow (longer recharge time), but you are interpreting them as being more connected to meteoric input and also are ignoring groundwater sources to recharge? That seems inconsistent to me.

This paragraph has been moved to the discussion and reformatted for clarity. While all borehole fluids most likely originate from a young meteoric groundwater source, we are interpreting Boreholes 6 + 8 as having a more direct source to surficial sources due to the lower metals concentrations and certain microbial community members that are shared with the soil samples. It is also difficult to discern true fluid flow in the subsurface at Edgar Mine due to a complicated fracture network that feeds the boreholes.

266: rainwater entering the system and reacting with the host rock sufficiently to achieve these chemical compositions, seems extremely rapid. As noted about, this is also not consistent with the microbiology. Please consider a groundwater source.

As we have mentioned in previous comments, the fluid source for the boreholes of Edgar Mine is likely of young meteoric groundwater origin. Within a geologic timeframe, this can be considered between tens to thousands of years, as the amount of water-rock interaction that would be required to influence the $\delta^{18}\text{O}$ and $\delta^2\text{H}$ values would require a great deal of time and alteration. We have made sure to address any miscommunication about the source of the borehole fluids being immediate meteoric water entering the subsurface, but actually a groundwater of meteoric origin that is relatively young.

289-290: I would suggest that DOC analysis would be a very useful means of tracking surficial inputs.

We do not have enough additional sample to measure DOC analysis. This could be followed up in a future study at Edgar mine.

292-293: please avoid hyperbole

We have removed the use of the word “recluse” from this line.

294: depleted suggests it was high and now it is low, do you know that?

We have addressed this word choice to reflect that the concentrations of inorganic nitrogen were low or below detection limits.

300-301: This argues against a very young meteoric origin and I am not sure that the cross plot is sufficiently useful to be a main figure in the paper.

We believe that the relationship between Mn and Zn is useful in determining metal-rich fluids and delineating between different borehole microbial community composition. The use of $\delta^{18}\text{O}$ and $\delta^2\text{H}$ values are for relative age approximations of the borehole fluids. While the borehole fluids may not be of very young meteoric origin, they represent a meteoric groundwater that reflects the initial conditions from surface recharge into the subsurface. It is clear that a longer residence time is required for the fluids to develop higher metals concentrations. Therefore, we believe this figure is useful for interpreting how the boreholes may have varying meteoric groundwater residence times to acquire different metals concentrations. It is important to note that the figure also demonstrate the conserved concentrations of the metals from the individual boreholes despite sampling period. Additionally, it serves as a useful tool for differentiating between different microbial communities within the subsurface.

368-369: I am not following this conclusion.

We have re-phrased this sentence for clarity.

[L 360-362 : Geochemical gradients determined by metals concentrations (by manganese and zinc) reflect the potential presence or absence of potential endemic subsurface microbial communities between borehole fluid samples.]

462: when were these particular holes drilled..

There is no recording of when the holes were drilled as they were a function of teaching demonstrations from Colorado School of Mines. We know the date our plug apparatus was installed into the borehole at the start of the experiment in 2016.

October 7, 2021

Dr. John R. Spear
Colorado School of Mines
Environmental Science and Engineering
1301 19th Street
Golden, CO 80401

Re: Spectrum00631-21R1 (Spatial and temporal constraints on the composition of microbial communities in subsurface boreholes of the Edgar Experimental Mine)

Dear Dr. John R. Spear:

Your manuscript has been accepted, and I am forwarding it to the ASM Journals Department for publication. You will be notified when your proofs are ready to be viewed. This is a cool study, well done and congratulations!

Sincerely,

Jeffrey Gralnick
Editor, Microbiology Spectrum

Journals Department
Supplemental Material File 2: Accept
Supplemental Material File 1: Accept